# RECAP: Recursive Context-Aware Reasoning and Planning for Large Language Model Agents

**Zhenyu Zhang[1]** [*][†]  **Tianyi Chen[1]**[*]  **Weiran Xu[1]**[*]  **Alex Pentland[2,3]**  **Jiaxin Pei[2]**

[1] Department of Computer Science, Stanford University
[2] Stanford Institute for Human-Centered AI       [3] MIT Media Lab

zhenyuz5@stanford.edu  tchen288@stanford.edu  weiran@stanford.edu
sandy@media.mit.edu  pedropei@stanford.edu

## Abstract

Long-horizon tasks requiring multi-step reasoning and dynamic re-planning remain challenging for large language models (LLMs). Sequential prompting methods are prone to context drift, loss of goal information, and recurrent failure cycles, while hierarchical prompting methods often weaken cross-level continuity or incur substantial runtime overhead. We introduce **ReCAP** (**Re**cursive **C**ontext-**A**ware Reasoning and **P**lanning), a hierarchical framework with shared context for reasoning and planning in LLMs. ReCAP combines three key mechanisms: (i) plan-ahead decomposition, in which the model generates a full subtask list, executes the first item, and refines the remainder; (ii) structured re-injection of parent plans, maintaining consistent multi-level context during recursive return; and (iii) memory-efficient execution, bounding the active prompt so costs scale linearly with task depth. Together these mechanisms align high-level goals with low-level actions, reduce redundant prompting, and preserve coherent context updates across recursion. Experiments demonstrate that ReCAP substantially improves subgoal alignment and success rates on various long-horizon reasoning benchmarks, achieving a 32% gain on synchronous Robotouille and a 29% improvement on asynchronous Robotouille under the strict pass@1 protocol.

## 1 Introduction

A fundamental characteristic of intelligence is the smooth transition between high-level abstract reasoning and low-level task execution—something humans routinely perform in everyday activities [14]. Imagine organizing a multi-day trip: one first outlines a broad plan—such as identifying destinations, transport, and accommodations—before refining it into actionable subtasks like booking tickets or arranging local travel. Real-world task execution, however, rarely follows a fixed script: a resource may be unavailable, an intermediate step may fail, or schedules may conflict. Such situations demand long-horizon reasoning that both preserves the global intent and maintains coherence between different levels of detail within the plan, while flexibly revising steps as feedback unfolds. Current LLM-based methods face challenges here: sequential prompting exhibits context drift and repeated cycles when early plans leave the context window, while hierarchical prompting often lose continuity across levels or incur redundant memory usage. Addressing such scenarios demands context-aware long-horizon adaptive reasoning and planning, requiring an intelligent system that can commit to a plan while remaining flexible to feedback and maintain coherence across multiple levels of reasoning [24, 5].

---

[*]Equal contribution.
[†]Project lead

39th Conference on Neural Information Processing Systems (NeurIPS 2025).

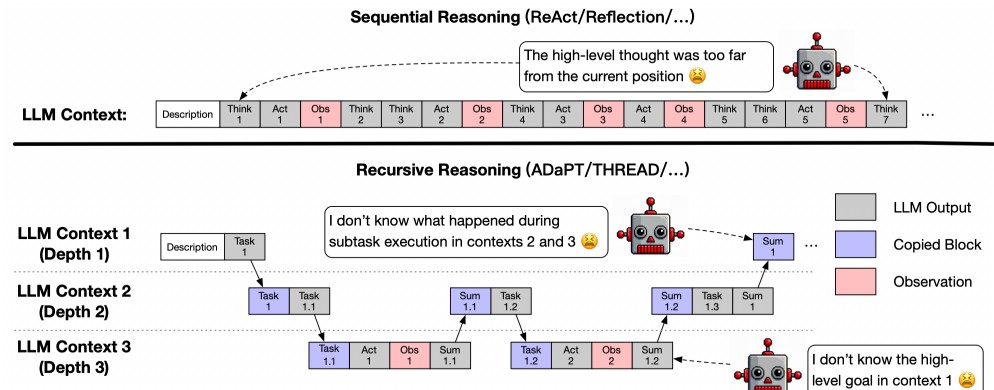

(a) **Limitations of pure sequential/hierarchical prompting.** In sequential prompting (e.g., ReAct/Re-flexion), early high-level thoughts drift far in a sequential history, leading to loss of goal information. In hierarchical prompting (e.g., ADaPT/THREAD), subtasks run in separate local contexts, fragmenting information across depths and leading to recurrent failure cycles.

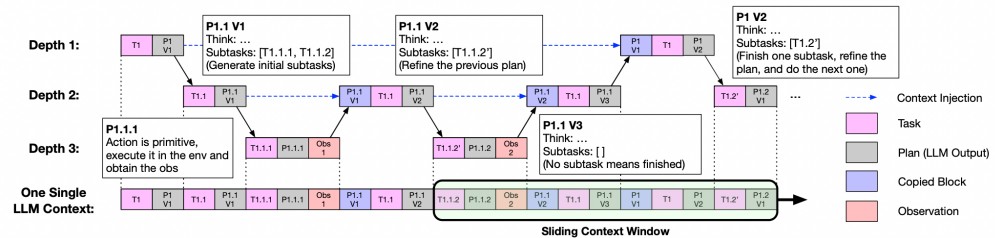

(b) **ReCAP.** At each depth, the agent generates or refines a subtask plan, executes primitive actions as they occur, and appends new observations. After resolving each subgoal, the parent's remaining plan is re-injected into a single sliding LLM context, keeping high-level intent proximal to the current decision point and preserving coherence across multiple levels of reasoning.

Figure 1: sequential/hierarchical prompting vs. RECAP

Recent reasoning frameworks enable LLMs to interleave reasoning with action, improving problem solving in multi-step settings. Sequential prompting methods such as Chain-of-Thought (CoT) [25], ReAct [5], and Reflexion [19] operate in a largely linear fashion: the model produces step-by-step thoughts (and optionally actions/observations) along a single trajectory. These approaches are simple and effective on short horizons, but in long-horizon environments, early plans often drift out of the context window or become stale, leading to loss of goal information and recurrent failure cycles even with extended contexts [11, 6, 2]. Ada-Planner [22] extends this paradigm by editing a single global linear plan through explicit closed-loop refinement, but it remains fundamentally linear and still susceptible to divergence from planned trajectory on long horizons. In response, hierarchical prompting methods explicitly organize reasoning beyond a single chain: Tree of Thoughts (ToT) explores branching and backtracking over thought trees [6]; Graph of Thoughts (GoT) generalize to non-linear dependencies [1]; THREAD recursively spawns subproblems but prompts each with isolated subgoal context [18]; ADaPT alternates planner–executor prompts to refine subgoals [16]; and REPL-Plan maintains an external program state and code-execution loop to drive planning [10]. While these methods organize reasoning in hierarchy, they can either (i) reduce continuity between levels of reasoning when subtasks are prompted in largely isolated contexts (e.g., THREAD, where subgoal prompts carry only partial parent information), or (ii) introduce increased runtime overhead and strong dependence on prior trajectories or tool-specific states (e.g., REPL-Plan, which introduces extra overhead by depending on an external code-execution environment).

To address these limitations, we introduce **ReCAP** (**Re**cursive **C**ontext-**A**ware Reasoning and **P**lanning), a hierarchical prompting framework with shared context for long-horizon tasks and dynamic environments. ReCAP is built around three major mechanisms. **(1) Plan-ahead task decomposition**: instead of generating one subtask at a time, ReCAP produces a complete subtask list in a single pass, executes only the first item, and refines the remaining plan upon subtask com-

pletion—preserving global intent while avoiding plan drift. **(2) Consistent multi-level context with structured injection**: reasoning across all recursion depths occurs within a shared LLM context. When returning from a subgoal, the parent's plan (latest thoughts and remaining subtasks) is re-injected into the active context window, preserving cross-level continuity and ensuring coherent progression through the task hierarchy **(3) Memory-efficient scalability**: the active prompt remains bounded, with critical planning information reintroduced through structured injection so that truncation does not cause loss of high-level intent. This avoids unbounded context accumulation, eliminates redundant few-shot duplication across recursive calls, and makes the external state grow linearly with recursion depth. Together, these mechanisms preserve high-level intent, support coherent multi-level reasoning, and provide robustness across a wide range of long-horizon tasks and dynamic environments.

We evaluate ReCAP across embodied and knowledge-intensive tasks with different planning horizons and feedback dynamics: **Robotouille** [7], **ALFWorld** [21], **FEVER** [23], and **SWE-bench** [9]. ALFWorld features short, largely linear embodied sequences. In Robotouille, the horizon grows much longer, and subgoals must be interleaved or refined continuously under resource contention. FEVER remains a shallow, tool-mediated retrieval task with a small symbolic action API. SWE-bench expands the action space from finite to effectively unbounded: the agent must compose multi-step code edits in a space far larger than environments with a limited verb set. Importantly, we adopt a strict **pass@1** protocol: each task instance is solved through a single uninterrupted reasoning–execution trajectory, without retries, self-consistency, or ensembling, and we use a one-shot demonstration per agent. This is stricter than the ReAct setting, which is frequently reported with pass@6 and multiple demonstrations, and thus better reflects realistic single-run agent deployment. To summarize, ReCAP introduces a recursive, context-aware framework for long-horizon reasoning and planning with LLMs. It enables structured subtask decomposition, dynamic memory tracking, and observation-driven plan adaptation without any training or fine-tuning. ReCAP outperforms strong baselines—achieving up to 32% in long-horizon tasks.

## 2   ReCAP: Recursive Context-Aware Reasoning and Planning

### 2.1   Framework Overview

Algorithm 1 formalizes the overall procedure, starting from the entry point $\texttt{ReCAP}(\langle D, O_{\text{init}}\rangle)$ where $D$ is the task description with one-shot demonstration and $O_{\text{init}}$ the initial observation. ReCAP frames model execution as a recursive process within a shared language model context $C$, which can be viewed as dynamically unfolding into a context tree: each recursive call extends LLM context $C$ with local reasoning traces and subtasks, while backtracking corresponds to returning control to the parent node. From $C$, the model first generates an initial thought $T$ and an ordered subtask list $S = \langle s_0, s_1, \ldots, s_{m-1}\rangle$ via the LLM-based planning function $\pi$. The node then advances by attempting the first subtask $S[0]$: if it is primitive, the environment dynamics $\mathcal{E}$ execute $S[0]$ and return an observation $O$, which is appended to $C$; if not, a recursive call is invoked on the extended context $C \parallel \langle T, S, S[0]\rangle$, after which the parent plan $\langle T, S[1:]\rangle$ is re-injected into $C$. In either case, the updated context is passed to the LLM-based refinement function $\rho$, which produces a revised thought and subtask list through prompt templates that adapt to different reasoning granularities (root, recursive, and backtracking levels; see Appendix D for full definitions). This loop continues until $S$ is empty or $\mathcal{E}$ reaches an end, yielding a complete resolution of the original task.

### 2.2   Recursive Task Decomposition with Plan-Ahead

ReCAP adopts a plan-ahead strategy: given the current context $C$, the planning function $\pi$ returns an internal thought and an ordered subtask list

$$(T, S = \langle s_0, s_1, \ldots, s_{m-1}\rangle) \leftarrow \pi(C).$$

At execution time, only the head subtask $S[0]$ is attempted; the rest $S[1:]$ is preserved for later refinement.

If $S[0]$ is itself a composite task, a recursive call is invoked on the extended context $C \parallel \langle T, S, S[0]\rangle$, producing a fresh thought and subtask list that decomposes the task to a lower level. This recursion continues until $S[0]$ is primitive—i.e., directly corresponds to an executable action under the current environment dynamics $\mathcal{E}$—at which point the subtask becomes a temporary leaf in the unfolding

**Algorithm 1** ReCAP

**Require:** LLM Context $C$
**Ensure:** Updated context $C$
 1: $(T, S) \leftarrow \pi(C)$
 2: **while** $S$ not empty **do**
 3:     **if** $S[0]$ is primitive **then**
 4:         $O \leftarrow \mathcal{E}(S[0])$
 5:         $C \leftarrow C \parallel \langle T, S, S[0], O \rangle$
 6:     **else**
 7:         $C \leftarrow \texttt{ReCAP}(C \parallel \langle T, S, S[0] \rangle)$
 8:     **end if**
 9:     $C \leftarrow C \parallel \langle T, S[1:] \rangle$
10:     $(T, S) \leftarrow \rho(C)$
11: **end while**
12: **return** $C$

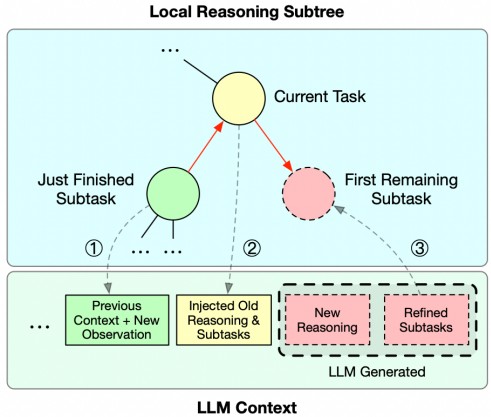

Figure 2: Overview of ReCAP's backtracking and refinement

hierarchy. Its execution may succeed or fail depending on environmental constraints. In either case, control returns to the parent context: the higher-level plan $\langle T, S[1:] \rangle$ is re-injected into $C$, and the refinement function $\rho$ is called to revise the subtask list.

## 2.3 Consistent Multi-level Context and Structured Injection

ReCAP maintains a shared LLM context across all recursion depths, so that high-level goals and low-level task executions remain aligned. When a non-primitive subtask $S[0]$ is encountered, the parent's plan $S$ is appended to the context before invoking a recursive call:

$$C \leftarrow \texttt{ReCAP}(C \parallel \langle T, S, S[0] \rangle).$$

After the subgoal is resolved, the parent's remaining plan $S[1:]$ is re-injected into $C$ to restore contextual awareness:

$$C \leftarrow C \parallel \langle T, S[1:] \rangle.$$

This re-injection step corresponds to backtracking: once a child subtask finishes (successfully or not), its outcome and the new environmental observations are appended to $C$, prompting the LLM to refine the parent's reasoning and update the remaining plan. By doing so, ReCAP dynamically prunes or revises subtasks in response to execution feedback, and then proceeds with the next unfinished subtask in the refined list.

Overall, this structured injection and backtracking mechanism helps the LLM to keep track of how its high-level plan got executed and encourages plan refinement throughout the long-horizon task execution. By keeping all reasoning within a single evolving context rather than allocating separate contexts at each level, ReCAP supports consistent context updates upon recursive return, makes use of history dialogue as context-aware demonstrations, and thus ensures coherent multi-level planning. Figure 2 illustrates the local structure of context re-injection and next subtask generation.

## 2.4 Sliding Window and Scalable Memory Efficiency

ReCAP is designed for bounded and efficient context usage. The active LLM prompt is limited by a sliding window of $K$ back-and-forth dialogue rounds (typically $K = 64$), with each round averaging $\bar{L}$ tokens. This keeps the active context size at $\mathcal{O}(K \cdot \bar{L})$, well within model capacities. Older rounds beyond the window are automatically removed, while critical planning information is reintroduced through structured context injection, so that truncation does not cause loss of high-level structure. Moreover, because all recursion operates within a shared context, few-shot examples are placed only once at initialization rather than re-injected at every recursive call. This significantly increases the proportion of tokens available for LLM reasoning. In contrast, prior recursive reasoning methods duplicate isolated contexts and few-shot prompts across calls, leading to fragmented usage. Although an external tree data structure is used to power ReCAP's structured injection, with each tree node holding $(T, S)$ for a subtask, once the subtask is refined, $(T, S)$ is replaced by the refined $(T', S')$. Therefore, at each step, only the path from the root to the current node remains active, so both the active prompt size and the external state storage scale as $\mathcal{O}(d \cdot \bar{L})$, where $d$ is the depth of the tree.

# 3 Evaluation

We evaluate ReCAP on four benchmarks, spanning coding, embodied, and text-based reasoning: ALFWorld [21], Robotouille [7], FEVER [23], and SWE-bench Verified [9]. All evaluations are conducted under a **pass@1** setting: each agent is allowed a single reasoning-execution trajectory per task instance, without retries, beam search, or sample-level ensembling. This setup is designed to highlight the raw decision-making capability of each agent to avoid any performance-enhancing strategies such as self-consistency, inner monologue [8], or response ensembling. ReCAP operates using one-shot prompting, with no training, offline or multi-trial optimization (e.g., Reflexion [19]), or task-specific tuning. To ensure consistency and fairness, for Robotouille, ALFWorld, and FEVER, we use the same task from the training sets to construct the one-shot examples and apply the same step limitations across agents, and use GPT-4o (2024-08-06) to conduct all the main experiments.

## 3.1 Benchmarks

**ALFWorld** is a symbolic, text-only household simulator for embodied reasoning via natural-language actions [21]. Each task (e.g., heat a tomato and place it on a dining table) is solved by composing short commands over a small action API (navigate, open/close, pickup/put, heat/cool, etc.). Task horizons are short (typically 5–15 steps), with limited interleaving of subgoals and modest long-range dependencies. We evaluate on the official unseen split with the provided symbolic interface. For few-shot construction, we pick one training task per each of the six categories (seen in prior work), adapt the narration to the agent's prompt format, and keep identical rule descriptions across agents. This setting probes whether ReCAP's structured backtracking still yields gains when horizons are shallow and the action space is small and discrete.

**Robotouille** is an embodied cooking environment designed to evaluate LLM agents on long-horizon tasks [7]. It supports two modes: *synchronous*, in which each action completes immediately and subgoals can be executed in a linear sequence; and *asynchronous*, where certain actions incur delays (e.g., cooking or waiting for water filling), allowing agents to interleave other subtasks during these waiting periods to mitigate temporal constraints and advance multiple goals concurrently. The required action sequence lengths display the complexity of the environment: synchronous tasks require between 10 and 57 steps, while asynchronous tasks range from 21 to 82 steps—even when measured using the minimal number of necessary actions. These lengths are substantially greater than those in ALFWorld, making Robotouille more challenging.

A major challenge in Robotouille is that subtasks often require multiple atomic actions to complete. For example, slicing a single vegetable requires three consecutive `cut` actions before the ingredient is marked as sliced. This means that even seemingly simple operations span multiple reasoning-execution steps, placing pressure on the LLM's context window. As subtask accumulates, early-stage plans and goals can be forgotten or overwritten by new observations. Additionally, the environment frequently introduces unexpected constraints: stations such as cutting boards may be occupied by other ingredients, forcing the agent to regenerate valid subtasks on the fly. We evaluate 10 synchronous and 10 asynchronous recipes, each with 10 official instances. For one-shot prompting, we follow prior work and use the held-out *onion–cheese sandwich* demonstration, adapting its wording to ReCAP's recursive format while preserving the original execution trajectory.

**FEVER** tests knowledge-intensive reasoning by classifying claims as `SUPPORTED`, `REFUTED`, or `NOT ENOUGH INFO` via a simplified Wikipedia API. Following ReAct, the agent uses three symbolic actions: `search[entity]` (first five sentences), `lookup[string]` (next sentence with keyword), and `finish[answer]` (final judgment). We use FEVER [23] to probe whether ReCAP's structured context management remains beneficial when horizons are short: most instances need less than 10 actions with little hierarchical decomposition. We evaluate 200 randomly sampled claims with a single shared demonstration from the training set adapted to each agent's prompt format.

**SWE-bench** is a benchmark dedicated to repository-level coding issue resolution. Agents take a GitHub issue as input and have access to a Docker container containing the repository. The agent must send commands to the container terminal to modify the code and pass a set of tests for each repository. In our work, we selected its human-validated subset, *SWE-bench Verified*, to evaluate our framework's ability to reliably solve real-world software issues [15].

Unlike embodied reasoning benchmarks such as ALFWorld [21], where the action space is concrete given the state of the environment, the action space of code editing is unbounded, since one can append, delete, or execute arbitrary code within the repository. Therefore, in ReCAP's evaluation on SWE-bench, there is no predefined list of valid actions. Instead, actions are interpreted solely from the language model's tool outputs, and any tool call accepted by the environment is considered valid. Furthermore, Since text outputs and tool-call outputs are separated in language model APIs, making the number of tool calls depend on the number of subtasks cannot be implemented without manipulating the probability space during decoding. In cases where a tool call is necessary to interact with the environment, we prioritize tool execution over subtask decomposition. ReCAP triggers a refinement once the tool call is executed and environment feedback has been received. To evaluate ReCAP, we modified SWE-agent's [26] memory from ReAct to ReCAP, added JSON schema and prompts for ReCAP output (see Appendix C.4, D.2), while keeping the tool execution and error handling the same. The LLM used is GPT-4.1 (2025-04-14), with no demonstrations and a temperature set to zero.

## 3.2 Baselines

**Sequential prompting baseline**  We retain the four sequential prompting baselines used in prior work: (1) **Standard**: directly output the full action sequence without intermediate thoughts/actions; (2) **CoT** [25]: add chain-of-thought to Standard; (3) **ReAct** [5]: interleave thoughts, actions, and observations; (4) **Act** (act-only prompting): remove thoughts from ReAct, akin to WebGPT's API-call style [13]. Their applicability follows the environment's interface: *Robotouille* exposes both the complete environment description and the currently executable actions, so we evaluate all four baselines. In contrast, *ALFWorld* provides only a partial environment description and the currently executable actions, so we evaluate Act and ReAct (Standard and CoT cannot bootstrap); *FEVER* follows the "search–lookup–finish" protocol from [5], so we evaluate all four (for Standard/CoT we prompt only the final verdict). For *SWE-bench Verified*, we used mini-SWE-agent [26] with GPT-4.1 (2025-04-14) as the ReAct baseline, which performance is publicly available.

**Hierarchical prompting baseline**  To complement missing embodied evaluations in the original papers, we additionally include the hierarchical prompting framework **ADaPT** [16] on Robotouille. We adapt its public prompting template to Robotouille's observation/action format while keeping its core control flow unchanged. As the method lacks an official Robotouille implementation, we follow its paper's prescribed prompts and heuristics, reporting our re-implementation under the same budget and limits as other agents. This addition allows a direct embodied comparison between sequential prompting (Act/ReAct/CoT/Standard), hierarchical prompting (ADaPT), and our multi-level context with localized backtracking (ReCAP).

**Excluded baselines**  We do not include **THREAD** [18], **Reflexion** [20], **Ada-Planner** [22], or **REPL-Plan** [10] in our main embodied comparisons for the following reasons. (i) **THREAD**'s uncontrollable recursion and its requirement for fine-grained, few-shot demonstrations for each subtask made it difficult to achieve reliable results within our experimental setup. (ii) **Reflexion** relies on multi-trial self-critique with an experience memory to improve over attempts; it is designed for pass@k or repeated episodes and is not well-aligned with our **pass@1** single-trajectory protocol. (iii) **Ada-Planner** edits a single global linear plan and depends heavily on code generation and an external interpreter/runtime. In our embodied settings, this introduces a toolchain mismatch; in practice, we observed frequent formatting and syntax errors that prevented stable execution under the same budget, making results non-comparable. (iv) **REPL-Plan** similarly depends on an external code-execution environment to maintain LLM-generated program states, adding substantial system complexity and biasing behavior toward reusing prior snippets, which makes it directly applicable to our prompt-only, tool-agnostic embodied setup. For fairness and reproducibility, we therefore focus on methods that operate within comparable interfaces and budgets.

Table 1: Pass@1 task success rates (%) across methods.

| Task | Step Range | ReCAP | ADaPT | ReAct | CoT | Act | Standard |
|---|---|---|---|---|---|---|---|
| ALFWorld | 4–25 | **91.0** | 71.6[3] | 84.0 | – | 74.0 | – |
| Robotouille (Sync) | 10–57 | **70.0** | 40.0 | 38.0 | 14.0 | 31.0 | 12.0 |
| Robotouille (Async) | 21–82 | **53.0** | 14.0 | 24.0 | 5.0 | 8.0 | 2.0 |
| FEVER | 1–10 | **63.5** | – | 63.5 | 58.5 | 58.5 | 53.5 |
| SWE-bench Verified [4] | 5-257 | **44.8** | – | 39.58 | – | – | – |

# 4 Results

## 4.1 Main Results

Table 1 shows that ReCAP performs better than ReAct on long-horizon tasks by a large margin. On synchronous Robotouille, it achieves a 32% gain, and on asynchronous Robotouille, a 29% improvement, representing its strength in long-horizon environments where early goals are prone to being overwritten in flat prompting setups. In these cooking tasks, high-level objectives often span multiple atomic actions and require multi-phase coordination. While ReAct performs all reasoning sequentially and suffers from context overflow, ReCAP maintains a dynamic task hierarchy and supports explicit backtracking, enabling the agent to preserve global goals and revise local decisions when needed. These features allow ReCAP to better handle the challenges of concurrent subgoals, delayed dependencies, and evolving observations, all of which are common in Robotouille.

We further perform a detailed failure case analysis to understand the nature of errors across different task difficulties in Robotouille. On **easy to medium tasks** (synchronous #1–5, asynchronous #1–3), ReCAP achieves near-perfect success, with failures mostly due to minor errors like missing the final cut in a sandwich or misplacing the completed item. In contrast, ReAct exhibits more fundamental mistakes even on simple recipes. On **long-chain or multi-dish tasks** (synchronous #8–10, asynchronous #4, #5, #8–10), ReCAP may occasionally make imperfect subtask choices, but it never enters an infinite loop. It consistently detects failure signals and backtracks to generate a revised plan. ReAct, on the other hand, frequently enters infinite loops when encountering blocked stations. For instance, if `lettuce1` is occupying `board2`, ReAct will repeatedly attempt to cut `onion1` by stacking and unstacking it on the blocking item without resolving the underlying issue. ReCAP avoids such failure by leveraging its multi-level context to identify the blockage and generate a corrected plan—e.g., moving `lettuce1` to an empty table before proceeding with the cut—thus maintaining task progress and preventing infinite loops. Figure 3 illustrates a detailed dialogue trace in which ReAct becomes trapped in an infinite unstack/stack loop, whereas ReCAP detects the looping behavior via backtracking, refines its plan to clear the blocked station, and then successfully resumes execution.

In shorter tasks, ReCAP's advantage becomes more nuanced. In ALFWorld, it improves over ReAct by 7%, a smaller but consistent gain. Although the reasoning horizon is shallower, we find that ReCAP is still able to recover from local errors. On FEVER, both methods perform similarly, achieving 63.5% accuracy. This result aligns with our expectation: when the number of reasoning steps is small, the benefits of explicit hierarchical structure and backtracking diminish. Nonetheless, the fact that ReCAP remains competitive without degrading performance demonstrates that our recursive architecture does not introduce overhead in short-horizon or knowledge-retrieval tasks while delivering increasing gains as task complexity and step length grow.

In real-world coding tasks, ReCAP achieves the highest success rate on SWE-bench Verified among all methods using GPT-4.1 as the LLM. All of the 500 tasks were successfully submitted without human intervention, of which 498 were successfully evaluated by the SWE-bench CLI and 224 tasks were solved. The horizon length and cost distributions are right-skewed, as shown in Figure 4. However, Figure 5 shows that the resolve rate does not drop entirely to zero for tasks involving

---

[3]ADaPT (ALFWorld) uses GPT-3.5. The ADaPT paper also reports GPT-4 runs, but the best accuracy is with GPT-3.5; we thus report GPT-3.5 here.

[4]`https://www.swebench.com/`, retrieved Sept 17, 2025

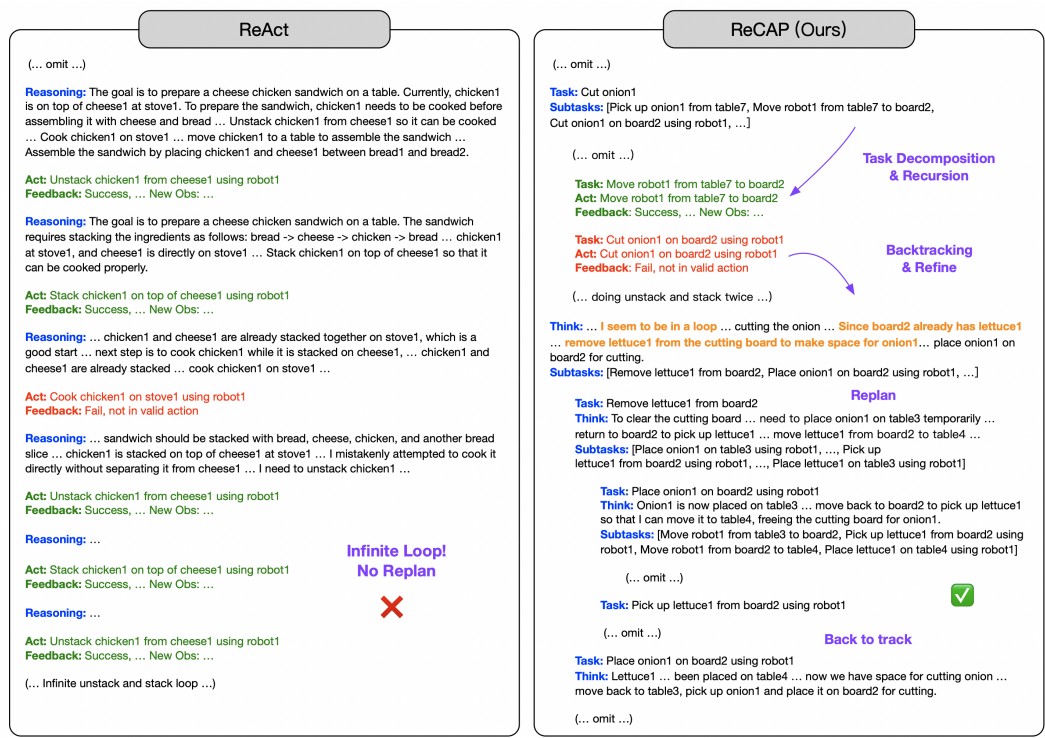

Figure 3: Detailed comparison between ReAct and ReCAP's behaviors when encountering blocked stations in Robotouille. **Left:** ReAct repeatedly alternates between stacking and unstacking the same item, resulting in an infinite loop. **Right:** ReCAP detects the loop, backtracks to clear the board by moving the blocking lettuce, and then proceeds with the correct sequence of actions.

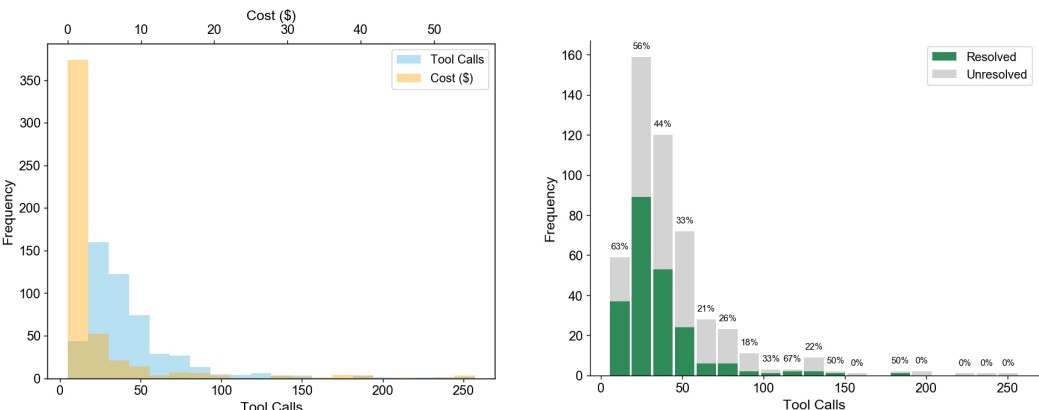

Figure 4: Tool call and cost distributions for ReCAP on SWE-bench Verified.

Figure 5: Task resolve rate of ReCAP on SWE-bench Verified, by number of tool calls.

more than 100 tool calls, implying that ReCAP indeed remains strong on tasks where it decides to run longer.

## 4.2 Task Tree Analysis

To understand the extent to which ReCAP decomposes a task before taking an action, we conducted an analysis on 20 runs of the Robotouille task `synchronous/6_lettuce_tomato_cheeseburger`. On average, the tree depth is 3.4 and the branching factor is 12.5, suggesting that ReCAP tends to produce shallow trees with moderate fan-out in embodied tasks, where each subtask often requires

multiple atomic actions to complete. For SWE-bench Verified, since the model's tool use takes higher priority than subtask decomposition, and each tool call incurs a return to the parent task for refinement, the resulting tree structure is similar to that of embodied tasks. A shallower tree also implies a smaller external state storage requirement.

## 4.3 Performance on Different Models

Table 2: Success rates (%) between ReAct and ReCAP across models, Robotouille tasks #2, 4, and 6.

| Method | GPT-4o | Qwen2.5-32B | Qwen2.5-72B | LLaMA-4 (400B) | DeepSeek-V3 (671B) |
|--------|--------|-------------|-------------|----------------|--------------------|
| ReAct  | 63.0   | 10.0        | 23.0        | 37.0           | 57.0               |
| ReCAP  | **90.0** | **33.0**  | **53.0**    | **60.0**       | **87.0**           |

To assess how our ReCAP generalizes across models compared to the standard ReAct setup, we evaluated both architectures on three representative synchronous Robotouille tasks (IDs 2, 4, and 6). To control API costs, we limited our experiments to these three tasks and imposed a hard cap of 64 context messages per run (any excess was truncated). We benchmarked four open-source and proprietary LLMs—Qwen2.5-32B [17], Qwen2.5-72B, LLaMA-4 (400B) [12], and DeepSeek-V3 (671B) [3]—alongside GPT-4o. Table 2 reports average success rates (%) for each model under both ReAct and ReCAP. ReCAP consistently outperforms ReAct across all evaluated models, demonstrating strong robustness and broad applicability. These results highlight the compatibility and reliability of our recursive framework across diverse model families, without requiring any model-specific tuning or code modification.

## 4.4 Ablation Studies

To further investigate the effectiveness of our structure, we conducted extended ablation studies on the Robotouille task `synchronous/6_lettuce_tomato_cheeseburger`, which requires 23 (theoretical optimal) to 40 (average) rounds of agent-environment interaction to complete. We also evaluate the statistical significance of differences in success rates between the structural variants and the original version. Table 3 reports the success rates and p-values for various ReCAP structural variants, including alterations to the maximum reasoning depth (**Level 2/3/4/5**), omission of reasoning traces during backtracking (**Name Only**), and modifying the output format to generate only decomposition/action outputs without the "think" reasoning (**No Think**), as well as passing all think history instead of just the most recent (**Think Many**). All structural variants were evaluated with a context length of 128, where context length refers to the number of messages stored in the LLM history during the conversation. For the **No Think** and **Name Only** variants, we adapted the one-shot prompt to match their structure.

Table 3: Success rates (%) between different ReCAP structural variants, Robotouille task # 6.

| Original | think_many | no_think | name_only | level_5 | level_4 | level_3 | level_2 |
|----------|-----------|----------|-----------|---------|---------|---------|---------|
| **80**   | 70        | 60       | 55        | 70      | 60      | 10      | 0       |

For the long-horizon task `synchronous/6_lettuce_tomato_cheeseburger`, the success rate degrades significantly when reasoning traces are removed or when the maximum reasoning depth is restricted (**Level 2/3**). This suggests that the explicit reasoning traces produced by ReCAP help the LLM perform better by allowing it to recall previous subtasks and lines of reasoning. With restricted reasoning depth, the LLM is limited in its ability to recursively decompose higher-level tasks into atomic, directly executable actions, forcing it to generate actions from insufficiently decomposed subtasks and thus reducing accuracy.

On the other hand, the **Think Many** and **No Think** variants achieve success rates comparable to the original, indicating that ReCAP is robust even when the LLM is provided with either excessive reasoning history or only decomposition/action outputs without the intermediate "think" reasoning. This robustness is also observed in the context length variants: no significant performance degradation

occurs when limiting the number of messages stored in the LLM history during the conversation, implying that ReCAP remains effective under context-sensitive scenarios.

### 4.5 Cost Estimate

We conducted cost estimation on Robotouille for ReCAP, and cost comparison between ReCAP and ReAct on ALFWorld. For the Robotouille task `synchronous/6_lettuce_tomato_cheeseburger`, the average number of LLM calls is $74.95$ with a standard deviation of $27.87$, and the average cumulative cost for one complete run is $7.77$ USD with a standard deviation of $3.45$ USD. For ALFWorld, the total cost of running all 134 tasks in the test set is $37.89$ USD using ReAct, and $118.40$ USD using ReCAP—approximately three times the cost of ReAct. We identified that the extra cost mainly comes from the additional reasoning traces in the input and the extra steps required for intermediate task decomposition.

## 5   Discussion

Despite its strong performance, ReCAP has several limitations. The framework delegates all decomposition, execution, and backtracking decisions to the underlying language model, without external validation or grounding. As a result, the system remains sensitive to model quality and may propagate errors if the LLM fails to follow instructions or misinterprets feedback. In addition, ReCAP's recursive design—while enhancing robustness and planning accuracy—incurs longer interaction trajectories compared to flat prompting. This leads to increased API overhead and slower end-to-end latency, which may pose challenges in deployment scenarios with strict efficiency or cost constraints.

Several promising directions remain for improving ReCAP. One is to modularize the architecture by decoupling high-level planning and low-level execution, allowing different models (e.g., a large LLM for decomposition and a lightweight model for primitive actions) to collaborate more efficiently. Another avenue is to reduce interaction costs through reasoning compression, dynamic step control, or API batching strategies. More broadly, ReCAP's recursive context tree challenges the default assumption that LLM context must be a linear dialogue history. Future work may explore structuring memory as an executable graph, enabling more targeted retrieval and potentially allowing reinforcement learning or memory-aware routing to optimize reasoning under context constraints. This perspective points toward a more scalable alternative to simply expanding context length—improving how context is organized and used, rather than how much is stored.

## 6   Conclusion

We introduce RECAP, a framework that makes long-horizon agents both coherent and adaptive by combining three simple but complementary ideas: (i) **plan-ahead task decomposition**, where the model proposes a full ordered subtask list, commits to the head item, and then refines the remainder using new observations to prevent myopic drift; (ii) **consistent multi-level context with structured injection**, which executes all recursion in a single LLM session and selectively re-inserts the parent's description, latest thoughts, and remaining subtasks on backtracking, preserving cross-level continuity without fragmenting context; and (iii) **sliding-window scalability**, which bounds the active prompt and re-surfaces only the essentials, so latency and memory scale with path depth rather than with total trajectory length. Across embodied cooking (Robotouille), symbolic interaction (ALFWorld/FEVER), and repository-level code editing (SWE-bench Verified), ReCAP improves success rates under a pass@1 protocol without training, fine-tuning, or tool-specific engineering, showing that how we organize and reinject context can matter as much as how much context we have.

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

# NeurIPS Paper Checklist

The checklist is designed to encourage best practices for responsible machine learning research, addressing issues of reproducibility, transparency, research ethics, and societal impact. Do not remove the checklist: **The papers not including the checklist will be desk rejected.** The checklist should follow the references and follow the (optional) supplemental material. The checklist does NOT count towards the page limit.

Please read the checklist guidelines carefully for information on how to answer these questions. For each question in the checklist:

- You should answer [Yes] , [No] , or [NA] .
- [NA] means either that the question is Not Applicable for that particular paper or the relevant information is Not Available.
- Please provide a short (1–2 sentence) justification right after your answer (even for NA).

**The checklist answers are an integral part of your paper submission.** They are visible to the reviewers, area chairs, senior area chairs, and ethics reviewers. You will be asked to also include it (after eventual revisions) with the final version of your paper, and its final version will be published with the paper.

The reviewers of your paper will be asked to use the checklist as one of the factors in their evaluation. While "[Yes] " is generally preferable to "[No] ", it is perfectly acceptable to answer "[No] " provided a proper justification is given (e.g., "error bars are not reported because it would be too computationally expensive" or "we were unable to find the license for the dataset we used"). In general, answering "[No] " or "[NA] " is not grounds for rejection. While the questions are phrased in a binary way, we acknowledge that the true answer is often more nuanced, so please just use your best judgment and write a justification to elaborate. All supporting evidence can appear either in the main paper or the supplemental material, provided in appendix. If you answer [Yes] to a question, in the justification please point to the section(s) where related material for the question can be found.

IMPORTANT, please:

- **Delete this instruction block, but keep the section heading "NeurIPS Paper Checklist",**
- **Keep the checklist subsection headings, questions/answers and guidelines below.**
- **Do not modify the questions and only use the provided macros for your answers**.


# A  Additional Framework Details

## A.1  LLM-call retry and history truncation

Our core API wrapper (`call_llm`) enforces two complementary safeguards:

**History-length cap:**   Before each call, if the in-memory history exceeds certain configurable entries, we delete the 2nd and 3rd elements (the oldest user–assistant exchange) while retaining the first entry (the fixed prompt containing few-shot examples and rules). Deletions always start from the second element to preserve the initial prompt (few-shot + rules).

**Model context-length cap:**   If the LLM returns a `context_length_exceeded` error, we delete entries 2 through 33 (i.e. the oldest 32 messages after the fixed prompt) and retry the same user prompt. In practice, with the history-length cap, it is exceedingly rare to hit the 128K-token limit; this branch exists as a fallback for extraordinarily long dialogues.

## A.2  Periodic rule reminder injection

To prevent "rule amnesia" over long trajectories, we prepend the full environment rule text every 10 LLM invocations. This ensures that the rules are reintroduced regularly, bolstering stability on multi-step tasks.

## A.3  Leaf-failure and nonleaf-failure prompt triggers

When the agent proposes an invalid primitive action (not in the current `valid_actions` list), we invoke `generate_leaf_up_fail_prompt` (See Section D.1.6), which supplies diagnostic hints and asks the model to repair its remaining subtasks. If a non-leaf node exhausts all subtasks without completion, we call `generate_nonleaf_judge_done_prompt` (See Section D.1.4) to have the model either confirm task completion or generate additional subtasks. These failure-driven backtracking prompts enable recovery from both execution errors and planning dead-ends.

# B  Per-task Result on Robotouille

Tables 4 and 5 report the per-task success rates for ReCAP and our five baselines on the 10 synchronous and 10 asynchronous Robotouille tasks. In the synchronous setting (Table 4), ReCAP achieves an overall average of 70%, compared to 38% for ReAct, with p-value $< 0.001$.

In the asynchronous setting (Table 5), ReCAP yields even larger gains. It improves average task success from 24% (ReAct) to 53%, with p-value $< 0.001$. These improvements imply ReCAP's strength in managing concurrent subtasks and adapting plans on the fly. Despite the inherent difficulty of tasks 9 and 10, where both methods struggle, ReCAP still achieves a non-zero success rate, while none of the baselines succeeded, underscoring the ability of recursive context tracking in long-horizon environments.

# C  Additional Experiment Details

## C.1  Robotouille Experiment Details

All Robotouille experiments use the same hyperparameters and evaluation protocol for both ReAct and other baselines. For each of the 10 synchronous and asynchronous test tasks, we run 10 random seeds under a pass@1 setting. We use the "onion cheese sandwich" recipe as our one-shot demonstration. The baseline agents employ the original few-shot prompt provided by the benchmark unchanged, while ReCAP uses that same example, adapted to emit a JSON-formatted output that contains its updated thinking and generated subtask list. Both methods share identical LLM settings (temperature = 0.5, default max tokens—GPT-4o's 128 K window—and no additional sampling penalties) and the same `max_step_multiplier=4` (up to four times the theoretical optimal step count per task) to allow room for error recovery. No other model-specific tuning is performed.

Table 4: Performance comparison between ReCAP and baselines on Robotouille synchronous tasks using GPT-4o.

| Task | ReCAP (%) | ReAct (%) | ADaPT (%) | Act (%) | CoT (%) | Std (%) |
|---|---|---|---|---|---|---|
| | **100** | 60 | 70 | 90 | 40 | 50 |
| | **80** | 30 | 70 | 30 | 30 | 20 |
| | **80** | 40 | 50 | 10 | 10 | 10 |
| | **80** | 60 | 50 | 30 | 10 | 0 |
| | **90** | 80 | 80 | 70 | 10 | 10 |
| | **70** | 60 | 60 | 30 | 30 | 30 |
| | **70** | 50 | 20 | 20 | 0 | 0 |
| | **60** | 0 | 0 | 0 | 0 | 0 |
| | **50** | 0 | 0 | 10 | 10 | 0 |
| | **20** | 0 | 0 | 20 | 0 | 0 |
| **Average** | **70** | 38 | 40 | 31 | 14 | 12 |

Table 5: Performance comparison between ReCAP and baselines on Robotouille asynchronous tasks using GPT-4o.

| Task | ReCAP (%) | ReAct (%) | ADaPT (%) | Act (%) | CoT (%) | Std (%) |
|---|---|---|---|---|---|---|
| | **50** | 20 | 20 | 20 | 10 | 10 |
| | **80** | 40 | 40 | 20 | 10 | 10 |
| | **90** | 10 | 10 | 0 | 20 | 0 |
| | **40** | 40 | 20 | 20 | 0 | 0 |
| | **50** | 10 | 0 | 0 | 10 | 0 |
| | **80** | 60 | 50 | 10 | 0 | 0 |
| | **70** | 30 | 0 | 0 | 0 | 0 |
| | **40** | 30 | 0 | 10 | 0 | 0 |
| | **20** | 0 | 0 | 0 | 0 | 0 |
| | **10** | 0 | 0 | 0 | 0 | 0 |
| **Average** | **53** | 24 | 14 | 8 | 5 | 2 |

## C.2 ALFWorld Experiment Details

All ALFWorld experiments follow the same LLM settings and evaluation protocol for both ReCAP and other baselines. We use the same model hyperparameters as described in Section C.1. The timeout action step is set to 50.

The original ReAct baseline employs a two-shot prompt for each task. For fairness, we only use the first example from the two-shot prompt for ReAct and modify it accordingly for ReCAP. The thinking and action traces are aligned between ReAct and ReCAP.

## C.3 FEVER Experiment Details

All FEVER experiments follow the same LLM settings and evaluation protocol for both ReCAP and other baselines. We use the same model hyperparameters as described in Section C.1, and we fix the random seed to 42 to shuffle the FEVER test set. From the shuffled pool, we sample 200 claims per run and evaluate under a hard cap of 10 reasoning steps (each agent may invoke at most 10 API calls—search, lookup, or finish). This configuration is applied to both ReAct and other baselines.

The original ReAct baseline employs a three-shot prompt; to isolate the effect of our recursive architecture, ReCAP uses only a one-shot prompt. When re-running the published baseline, we discovered a typo in its few-shot examples: the label "NOT ENOUGH INFORMATION" was used instead of the official "NOT ENOUGH INFO," causing misclassification and an artificial drop in accuracy. We corrected this label in our reproduced baseline to ensure a fair comparison.

## C.4 SWE-bench Verified Experiment Details

All SWE-bench experiments were conducted on 500 tasks under the pass@1 setting using GPT-4.1, with the temperature fixed at 0 and no demonstrations provided. We modified the original SWE-agent code, which uses LiteLLM to call the OpenAI API. All experiments were run without any human intervention (e.g., terminating runs that took too long). To encourage proper task submission, we added an extra parent node to the task tree. When the ReCAP agent successfully solves a task, it returns to this parent node (named "review and submit") to trigger submission.

Since no few-shot demonstrations were used, we employed a JSON schema to constrain the LLM's output to the fields "think" and "subtasks." The JSON schema is provided below:

```
"schema": {
    "type": "object",
    "properties": {
        "think": {
            "type": "string",
            "description": "Your reasoning and thought process
                for the current task."
        },
        "subtasks": {
            "type": "array",
            "items": {
                "type": "string"
            },
            "description": "A list of subtasks to complete the task.
                Empty if you determine the task has been completed.
                Do not generate any subtasks beyond the scope of the
                current task."
        }
    },
    "required": ["think", "subtasks"],
    "additionalProperties": False
}
```

For the baseline comparison, we used the accuracy reported by mini-SWE-agent on the benchmark[5]. Upon examining both their code and paper [26], we confirmed that mini-SWE-agent adopts the ReAct framework.

# D Prompts

Here we list all of the prompts used in our framework, such as those driving the recursive decomposition, backtracking refinement, and subtask execution, as well as the rule templates for each of the three benchmarks. Due to their length, we have omitted the few-shot demonstration prompts for each benchmark from this Appendix; readers can refer to our public code repository for the complete examples.

## D.1 Core Framework Prompts

Below we list all of the prompt templates used by ReCAP in the recursive decomposition and backtracking loop. Variable placeholders (e.g. '{task_name}') indicate where runtime values are substituted.

### D.1.1 Initial Decomposition Prompt

This prompt starts the process by providing system instructions, environment rules, initial observation, and the high-level goal.

```
{system_prompt}

Here's the rule of the environment:
{rule}

{init_obs}

Now you need to find the answer for the following question using the
actions I provide.
Here is the description:
{task_name}

Now, start the task. Please firstly generate a list of general subtasks to
accomplish the task.
```

### D.1.2 Recursive (Downward) Prompt

Used when descending into a newly created subtask node to request its own subtasks.

```
OK.

Your current task: {task_name}

We wish you to generate a list of subtasks for the current task.
```

### D.1.3 Leaf Backtracking Prompt

Triggered after successfully executing a leaf subtask when the parent still has remaining subtasks.

```
You have completed the task: {done_task_name}

Here is the result:
{obs}

Now, you return to the parent task:
```

---
[5]https://www.swebench.com/, retrieved Sept 17, 2025

```
Your current task: {previous_stage_task_name}

Your previous think: {previous_stage_think}

Your remaining subtasks:
{remaining_subtask_str}

We wish you to refine your list of subtasks based on the latest observation
to achieve your goal.
If there are no remaining subtasks, check if the goal is achieved.
If yes, return an empty list; otherwise, return the required subtasks.
Do not generate subtasks beyond the current task.
```

### D.1.4 Non-Leaf Completion Prompt

Fires when a non-leaf node has no subtasks left and needs to decide if it's fully done or generate more.

```
You have successfully completed the task: {done_task_name}

Now, you return to the previous stage.
Your current task: {previous_stage_task_name}

Your previous think: {previous_stage_think}

There are no remaining subtasks. Determine if the task is complete.
If it is, set subtasks to an empty list; if not, generate necessary
subtasks.
```

### D.1.5 Leaf Completion Prompt

Used when a leaf node ends with no subtasks remaining to perform the final completion check.

```
You have completed the task: {done_task_name}

Here is the result:
{obs}

Now, you return to the previous stage.
Your current task: {previous_stage_task_name}

Your previous think: {previous_stage_think}

There are no remaining subtasks. Determine if the task is complete.
If it is, set subtasks to an empty list; if not, generate necessary
subtasks.
```

### D.1.6 Leaf Failure Prompt

Triggered when a leaf action fails validity, requiring the parent to fix the subtasks.

```
You fail to complete the task: {fail_task_name}
Because the action is not among the valid options.

{obs}

Now, you return to the previous stage.
Your current task: {previous_stage_task_name}

Your previous think: {previous_stage_think}
```

```
Your remaining subtasks:
{remaining_subtask_str}

We wish you to modify your subtasks to fix the error.
```

### D.1.7 Non-Leaf Backtracking Prompt

Issued after any non-leaf child finishes, to refine the parent's remaining subtasks.

```
You have completed the task: {done_task_name}

The result shows in the previous context.

Now, you return to the parent task:
Your current task: {previous_stage_task_name}

Your previous think: {previous_stage_think}

Your remaining subtasks:
{remaining_subtask_str}

We wish you to refine your list of subtasks based on the latest observation
to achieve your goal.
Do not generate subtasks beyond the current task.
```

### D.2 Tool Call Prioritized Prompt for SWE-bench

#### D.2.1 Action (Tool Call) Taken

```
Latest observation:
{obs}

Your current task: {task_name}

Your previously proposed subtasks:
{remaining_subtask_str}

You can refine the subtasks based on the latest observation, empty list if
you think your current task is done, further decompose the first subtask by
not calling any tool, and/or make tool calls to make progress.
```

#### D.2.2 Recursive (Downward) Prompt

```
OK.

Your current task: {task_name}

You can decompose the task if it is too complex, empty list if you think
your current task is done, further decompose the subtask by not calling any
tool, and/or make tool calls to make progress.
```

#### D.2.3 Backtracking (Upward) Prompt

```
You have determined that the task {done_task_name} has been completed.

Now, you return to the parent task.
Your current task: {previous_stage_task_name}

Your previous think: {previous_stage_think}
```

Your remaining subtasks:
{remaining_subtask_str}

You can refine the subtasks based on the latest observation, empty list if
you think your current task is done, further decompose the first subtask by
not calling any tool, and/or make tool calls to make progress.

## D.3 Rule Templates

### D.3.1 Robotouille

You are an agent exploring a game environment with a goal to achieve. You
will propose a series of task decomposition or an action in the current
state to make progress towards the goal. Follow the rules carefully since
the environment may have constraints that do not align with the real world.

You must propose a series of task decomposition or an action given the
current observation and valid actions. I will help you keep track of your
progress by providing your previous thoughts and remaining subtasks.

You will receive the initial state and the goal as follows:
Optional[Error Feedback: ...]
Observation: ...
Valid Actions: ...

where
- 'Observation' contains state information about objects in the environment
and the goal
- 'Valid Actions' is the list of actions you can take in the current state
- 'Error Feedback' includes feedback about an invalid action taken in a
previous interaction (not included in the history)
- This feedback is automated and shows if the action is either
syntactically incorrect or does not exist in the valid actions list
- This feedback does not check for semantic correctness and should neither
reinforce nor discourage the current strategy

Always format your response in json format:
{
"think": "",  // str: Your thought
"subtasks": [...],  // List[str]: The updated subtask list for completing
the task. If your current task can be executed directly from valid actions,
the list should include only that action. If the task is done, i.e. no
remaining subtasks, the updated subtask list should be empty.
}

Below is a description of the environment:
You are a robot in a kitchen environment. The objects in the kitchen and
your goal are described
in the Observation. The various types of objects in the kitchen include
- Station: A location in the kitchen where you can perform special actions,
e.g. cooking or cutting
- Item: An object that can be picked up and potentially used in a Station
- Player: Robots, including you, that are present in the kitchen
- Container: An object that can hold meals, e.g. a pot or a pan
- Meal: A mixture of ingredients contained within a Container

The rules of the environment are as follows:
- A Player can only hold a single Item at a time

- An Item must be placed on a Station to perform an action on it
- A Station must contain a single Item to perform an action on it
- Items can only be stacked on top of one another, but not underneath
- A Container must contain a Meal to have items added to it
- A Meal can be transferred between Containers
- When you cut an item, you must cut 3 times in succession, not immediate
- You can't place an object directly on a Station/Board when it's occupied
by another object. It's impossible to stack two items side by side on
the same Station/Board. Otherwise you will stack on it. You might want to
remove the object underneath first.
- If you cook an item, it is fully cooked after 3 timesteps, not immediate.
This is asynchronous (only consumes one action)
- If you fry an item, it is fully fried after 3 timesteps, not immediate.
This is asynchronous (only consumes one action)
- If you boil an item, it is fully boiled after 3 timesteps, not immediate.
This is asynchronous (only consumes one action)
- If there's nothing else you can do while waiting for an item to finish
cooking / frying / boiling, you have to generate n * 'Do Nothing' actions,
n =
the number of timesteps remaining until the item is finished cooking /
frying / boiling

Follow this recipe guide to learn how to make food in Robotouille:
Sandwich - On a empty table, put a slice of bread, stacked on prepared
ingredients, stacked on another slice of bread. The bread must directly
touch the table.
Hamburger - On a empty table, A bottom bun, stacked on prepared
ingredients, stacked on a top bun. The bottom bun must directly touch the
table.
Soup - A pot is first filled with water, then boiled while ingredients are
added, then served in a bowl when ready.

### D.3.2   ALFWorld

You can only pick up one item at a time. Always use Inventory to check what
you have in possession when you are not sure or plan to pick up an item.

### D.3.3   FEVER

You must propose a series of task decomposition or an action given the
current observation and valid actions. I will help you keep track of your
progress by providing your previous thoughts and remaining subtasks.

You will receive the initial state and the goal as follows:
Observation: ...
Claim: <some factual statement to verify>

Solve a claim-verification task with interleaving Thought, Action,
Observation steps.
- Thought can reason about what you know so far and what you need to check
next.
- Action must be one of:
  1. Search[Entity] - searches exactly that Wikipedia entry and returns the
first paragraph if found; if not, returns a list of similar titles to try.
  2. Lookup[string] - returns the next sentence in the current passage
containing that exact string.
  3. Finish[answer] - ends the task by outputting one of: SUPPORTS,
REFUTES, or NOT ENOUGH INFO.

```
Some important reminders:
- Only use Finish[answer] when you're confident no further lookup or search
can change your decision.

Always format your response in json format:
{
    "think": "",
    "subtasks": [...],
}
```

# E    Social Impacts

ReCAP is a general-purpose framework designed to enhance the long-horizon reasoning and planning capabilities of language model agents. Its potential positive societal impacts span multiple domains: in education, ReCAP can enable tutoring agents that adaptively break down complex learning goals into structured steps; in assistive robotics, it may support more reliable household or caregiving robots that execute long-horizon tasks with contextual awareness; and in scientific discovery, it could help automate complex experimental or literature-based reasoning pipelines.

While ReCAP introduces no new content generation capabilities, it enhances the decision-making structure of existing language models. As such, it does not pose significant new risks related to misinformation, bias, or autonomy beyond those already present in the underlying LLMs. Nevertheless, any increase in agent autonomy warrants careful consideration. In applications with safety-critical consequences—such as healthcare, law, or finance—explicit validation, oversight, and transparency mechanisms should be incorporated before deployment.

We encourage future work to explore safeguards, such as interpretable reasoning trees and confidence-aware action proposals, to ensure that recursive planning frameworks like ReCAP remain aligned with human oversight and accountability. Overall, we believe ReCAP promotes safer long-term reasoning in LLM agents and opens promising directions for building trustworthy AI systems.

