# OpenReview forum: "ReCAP: Recursive Context-Aware Reasoning and Planning for Large Language Model Agents"
_NeurIPS.cc/2025/Conference — NeurIPS 2025 poster_

### Official Review · Reviewer_qEKq · 2025-06-23

**Clarity:** 3
**Significance:** 2
**Originality:** 2
**Rating:** 4
**Confidence:** 4

**Summary:**

This paper presents a novel approach for real-time planning with LLMs. The novelty relies on the use of an LLM to decompose problems into subtasks recursively until reaching primitive actions, while maintaining the decomposition tree with contextual information to interleave planning and execution.

The approach is tested over three problems, Robotouille, ALFworld and FEVER, and it is shown to improve performance over problems with long planning requirements and a concurrent goals.

**Questions:**

Is there any problem that requires interleaving tasks, where completing one task would make it impossible to complete the other? This can be illustrated by the Sussman Anomaly problem.

Is there any problem in the experiments where a deadlock can occur, as in, the problem becomes unsolvable due to the execution of an action?

Can you give insights into the size of the context tree and number of backtracks?

Can you give insights into the runtime overhead and the increased number of LLM calls with respect to the baselines?

Given that the aim is to have real-time reasoning, how long does it take to return an action? What is the impact of different recursion levels in terms of time?

**Ethical Concerns:**

["NO or VERY MINOR ethics concerns only"]

**Final Justification:**

After the discussion with authors I'm confident that they can address the issues in the discussion, hence I raise my rating to 4.

**Quality:**

2

**Strengths And Weaknesses:**

# Quality

The paper provides a sound algorithm describing the high level procedure followed to recursively decompose the problem and maintain the relevant data in each node. The experimental design is clearly presented, and the results are discussed in detail.

That said, a few experiments were missing in order to understand the complexity of the tasks and the context tree. No results are provided in terms of the number of backtracks, the number of LLM calls, or the size of the resulting context trees as a function of plan legth. See questions below.

# Clarity

The paper is accessible and easy to follow at a high level. Certain sections would require more formal or precise definitions. For example, the setup for interleaving planning and execution is not clearly defined. I couldn’t understand how much time is allowed for each phase, or if this depends on the duration of the action execution. I couldn’t tell either if backtracking meant real backtracking in the environment, or backtracking  as in changing the chosen subtask or planned next action. This became clear in the experimental section, but not while reading about the method. In general, I would recommend adding text about how the problems are described and how prompts are designed.

In general, superlatives can be misleading. Stating that the results of LLMs using ReAct are “impressive” (line 26), over complex sequential reasoning is not well defined and contradictory. It is contradictory because the experimental results in this paper show that ReAct is not impressive over tasks with more than 10 actions. It is not clear either with respect to which measure or other computational approaches is the performance impressive. The adjective “complex” is ambiguous at this stage of the paper. What makes a problem to be “complex” is not a trivial statement.

Minor comments:

- Figure 1 is not referenced in the text, and it is not self-explainable. I don’t understand what the colours mean.
- Add reference to the algorithm lines when describing it in section 2.3 and 2.4.
- One shot example is referenced for the first time in line 128, but no context is provided to understand how is it used. This is partially due to the fact that the prompt structure is not provided.
- extra dot (line 171)
- Line 197-198 is repeated (see line 131)
- Figure 3 does not add much with respect to the textual explanation, which does a great job already. It can be removed to make space for some of the experiments needed to answer the questions below.
- It is unclear what does stale reasoning means (line 264).

# Significance

The results are significantly better than baselines over long horizon planning, clearly showing the potential of representing the context as a recursive tree, as well as providing the right level of detailed context.

# Originality

The paper provides a valuable insight of treating context as a non-linear data-structure and showing how it can improve the reasoning ability for long-term planning that interleaves reasoning and execution. The only caveat is that the paper needs better positioning with respect to work on hierarchical task decomposition in planning, e.g. Hierarchical Task Networks (HTN), and clarify how it differs from HTNs. The first claim of novelty should be clarified that it is in the context of ReAct, but that recursive hierarchical decompositions has a large body of research. The problem of interleaving reasoning and acting could also be placed in the broader context beyond the approaches used by LLMs (line 286). See work in “Ghallab, Malik, Dana Nau, and Paolo Traverso. Automated planning and acting. Cambridge University Press, 2016.”

---

> ### Author Rebuttal · Authors · 2025-07-31
>
> We thank the reviewer for the valuable feedback and thoughtful suggestions. Please find our detailed responses below:
>
> ---
>
> ### Terminology Clarifications: “Complex” and “Impressive”
>
> We agree that certain terms (e.g., “complex” and “impressive”) could benefit from more precise framing.
>
> - **On “impressive” (line 26):**
>   Our intention was to acknowledge prior work such as ReAct and Reflexion, which have demonstrated strong performance on short-horizon tasks. However, we agree that this adjective can be misleading in light of our results. We will revise this statement to focus on specific capabilities (e.g., short-term planning, interleaving reasoning with action) rather than making broad claims about performance.
>
> - **On “complex” tasks:**
>   We recognize that “complex” is a context-dependent term. In our paper, we primarily refer to long-horizon, multi-stage, and feedback-driven tasks—particularly those requiring more than 10 interleaved reasoning–execution steps—as “complex.” We will clarify this definition in the introduction. To support this empirically, we provide per-task breakdowns of ReAct’s failure modes in Appendix B, demonstrating that its success rate drops significantly on Robotouille tasks with longer step ranges or asynchronous subgoals.
>
> - **On the planning/execution interface and backtracking:**
>   Thank you for highlighting this point. Backtracking in ReCAP is not a physical rollback in the environment, but rather refers to revisiting the parent subtask with updated observations and regenerating the remaining plan. The environment only evolves forward. We will clarify this distinction in Section 3.2 and explicitly define the interleaving behavior in our setup description.
>
> - **On prompt construction and task goal encoding:**
>   We appreciate the suggestion to improve our description of prompt construction and goal encoding. We include concrete prompt templates and examples in Appendix E, and will reference these earlier in the main text to aid clarity.
>
> ---
>
> ### Interleaving and Conflicting Subgoals (Sussman Anomaly)
>
> **Q: Is there any problem that requires interleaving tasks, where completing one task would make it impossible to complete the other?**
>
> Yes, interleaving tasks is necessary in problems where subgoal interactions can lead to conflicts if executed in isolation. ReCAP handles such cases via recursive execution and its dynamic context tree. After each subtask execution, ReCAP backtracks to the parent node, which re-evaluates the subgoal structure using updated environment observations and prior reasoning stored in the context tree. This mechanism allows the system to detect when earlier decisions may hinder future progress and refine the subtask plan accordingly—enabling the interleaving of dependent tasks without redundant trial-and-error.
>
> ---
>
> ### Deadlocks and Irreversible Actions
>
> **Q: Is there any problem in the experiments where a deadlock can occur, i.e., the problem becomes unsolvable due to the execution of an action?**
>
> Yes, this can occur in Robotouille, particularly in asynchronous tasks where certain resources (e.g., cutting boards, tools, ingredients) can be occupied or transformed in ways that block progress. For instance, placing an ingredient on an already-occupied tool or prematurely modifying an item (e.g., slicing before transferring) can make it temporarily unusable or incompatible with later subgoals, potentially leading to deadlocks in ReAct planners. Such failure modes are frequently observed in ReAct (see Figure 3 and Section 4.1), where agents enter infinite action loops when failing to account for dynamic environment constraints.
>
> ReCAP avoids such deadlocks by continually re-evaluating the plan via backtracking. After each subtask execution, it revisits the parent node with updated observations and reasoning, allowing it to detect unintended consequences and revise the subtask tree accordingly. This mechanism enables ReCAP to recover from potential deadlock situations that would otherwise halt sequential planners.
>
> ---
>
> ### Context Tree Size and Backtracking
>
> **Q: Can you give insights into the size of the context tree and number of backtracks?**
>
> The size of the context tree varies with task complexity and execution dynamics. For medium-complexity Robotouille tasks (e.g., asynchronous task #7), we observe an average of 100–120 nodes per trajectory, with approximately 40 leaf nodes (i.e., primitive executable actions). The remaining nodes correspond to higher-level subtasks generated via recursive decomposition. As ReCAP traverses the context tree in a depth-first fashion, the number of backtracks is equal to the number of nodes minus one. We appreciate the suggestion and will consider including more detailed statistics on tree structure and backtracking dynamics in future revisions.
>
> ---
>
> ### Runtime Overhead and LLM Call Volume
>
> **Q: Can you give insights into the runtime overhead and increased number of LLM calls compared to the baselines? What is the impact of recursion levels in terms of time?**
>
> ReCAP introduces additional LLM calls due to its recursive decomposition and backtracking mechanism, but this overhead is necessary to support context-aware reasoning in long-horizon settings.
>
> - For a representative Robotouille task (e.g., synchronous/6_lettuce_tomato_cheeseburger), ReCAP issues an average of ~75 LLM calls per episode (std ≈ 28), compared to ~20–25 for ReAct. The increased call volume arises from ReCAP's subgoal-level prompts and backtracking updates, which improve execution reliability and success rates.
>
> - **Latency:** Each LLM call takes 1–2 seconds on GPT-4o, depending on prompt length and model load. Thus, the time to return a single action is typically 2–3 seconds in practice. With ~75 LLM calls, this translates to a total runtime of approximately 2.5–4 minutes per episode.
>
> - **Recursion Depth:** Tasks with recursion depth up to 3–4 (which is sufficient for most cases) complete within reasonable real-time latency bounds. Deeper recursions (5+) offer marginal performance gains but increase delay, suggesting that shallow recursion achieves a good balance between speed and performance.
>
> To further support real-time applications, we plan to explore:
> - Batching multiple subtasks per call when possible
> - Model specialization, where a small, fast model executes leaf nodes
> - Prompt compression and dynamic context pruning to reduce token-level latency
>
> ---

---

> > ### Comment · Reviewer_qEKq · 2025-08-04
> > **Discussion**
> >
> > Thank you for your response!
> >
> > Can you clarify further the explanation with respect to the number of backtracks? To get a sense of the underlying context-tree, it would be helpful to understand the average branching factor and depth of the tree. If you have a DFS traversal over 120 nodes, it doesn't mean that you have 119 backtracks. I may be misunderstanding your explanation.
> >
> > I'd also appreciate more details on how the context tree would look like for the Sussman anomaly task, e.g. what would be the subtasks and what would be the decomposition? I'm not sure if it's feasible under a short time request, but a trace like figure 3 would be great.

---

> ### Author Response · Authors · 2025-08-05
>
> Thank you for your thoughtful follow-up. Below, we address your questions regarding backtracking, the tree structure, the Sussman anomaly trace, and the prompt. We also include an additional clarification regarding our relation to Hierarchical Task Networks (HTNs).
>
> ---
>
> ### On Clarification of Backtracks
> In ReCAP, we define a **backtrack** as the event that occurs whenever a leaf or intermediate subtask completes—regardless of success or failure—and the LLM returns to the parent node to refine its plan (see Section 2.4 and Figure 2). Conceptually, every time a child node completes, the system backtracks to its parent, incorporates the new observation and the completed subtask into the context, and then replans the next steps.
>
> ---
>
> ### Tree Structure Statistics
>
> We analyzed 20 `Robotouille/synchronous/6_lettuce_tomato_cheeseburger` traces:
>
> - **Avg. Tree Depth**: 3.4
> - **Avg. Branching Factor**: 12.5
>
> This suggests ReCAP tends toward shallow trees with moderate fan-out.
>
> ---
>
> ### ReCAP Trace on a Sussman Anomaly–Style Task
>
> Below is a trace on a Sussman anomaly–like scenario in *Robotouille*, where the **top bun** and **bottom bun** are stacked together, and the **patty** is on a different table.
>
> #### **Initial State**
> - `table1`: contains `bottombun1` (bottom), `topbun1` (top)
> - `table2`: empty
> - `table3`: contains cooked `patty1`
> - `robot1`: at table1, holding nothing
> - **Goal**: Assemble `bottombun1 → patty1 → topbun1` as a burger
>
>
> > _(… omit …)_
> >
> > **Task:** Prepare a hamburger on a table
> > **Subtasks:**
> > 1) Unstack **topbun1** from **bottombun1**
> > 2) Move **bottombun1** to an empty table
> > 3) Move **patty1** and stack on **bottombun1**
> > 4) Stack **topbun1** on **patty1**
> >
> > **Act:** Unstack **topbun1** from **bottombun1** using robot1
> > **Feedback:** ✅ Success — robot1 now holds **topbun1**.
> >
> > ---
> >
> > ### Backtracking & Refine
> > **Task:** Prepare a hamburger on a table
> > **Think:** With **topbun1** in hand, park it first, then assemble.
> > **Subtasks (refined):**
> > 1) Place **topbun1** on **table2**
> > 2) Fetch/move base and patty → finish stack
> >
> > **Act:** Move robot1 to **table2** → Place **topbun1** on **table2**
> > **Feedback:** ✅ Success — **topbun1** is now on **table2**.
> >
> > ---
> > ### Subtask Blocks Goal → Replan
> > **Task:** Fetch/move base and patty
> > **Think:** Gather all ingredients on **table2** and stack
> > _(… omit …)_
> > **Act (attempt):** Place **bottombun1** directly on **table2**
> > **Feedback:** ❌ Fail — **table2** is occupied by **topbun1** (cannot place two items side-by-side).
> > **Replan:** I cannot place **bottombun1** directly on **table2** because it is occupied by **topbun1**, ... Instead, I need to move to **table3** or **table1**, place **bottombun1** there (since they are unoccupied), and move **patty1** to the same table for stacking with buns. A possible solution is to move **bottombun1** to **table1** first.
> > **Subtasks (replanned):**
> > 1) Move to **table1** → place **bottombun1** on **table1**
> > 2) Go to **table3** → pick up **patty1** → return to **table1** → stack **patty1** on **bottombun1**
> > 3) Go to **table2** → pick up **topbun1** → return to **table1** → stack **topbun1** on **patty1**
> >
> > **Act:** Place **bottombun1** on **table1**
> > **Feedback:** ✅ Success — **bottombun1** is base on **table1**.
> >
> > **Act:** Pick up **patty1** at **table3** → move to **table1** → stack **patty1** on **bottombun1**
> > **Feedback:** ✅ Success — **patty1** now directly on **bottombun1**.
> >
> > **Back to Track (finalization)**
> > **Act:** Pick up **topbun1** at **table2** → move to **table1** → stack **topbun1** on **patty1**
> > **Feedback:** ✅ Hamburger completed on **table1**.
> > _(… omit …)_
>
> ---
>
> ### Response to Comment on HTN Positioning
> While ReCAP shares structural inspiration with HTNs, it differs fundamentally in execution. Classical HTNs require predefined task schemas and static method libraries, whereas ReCAP dynamically generates and revises subtasks via LLMs, without any predefined templates. Subtask plans are updated on the fly based on real-time observations and execution outcomes. This makes ReCAP more suitable for open-ended, feedback-driven environments. We will clarify this distinction and cite Ghallab et al. (2016) in the revised version.
>
> ---
>
> ### About Prompt
> Due to space limitations, we include the full one-shot prompt example in the supplementary material, while the prompt structure is detailed in Appendix E. We will revise the main text to include appropriate references to both the appendix and the supplementary material when introducing the one-shot example.

---

> > ### Comment · Reviewer_qEKq · 2025-08-05
> > **Discussion**
> >
> > Thanks for the clarification regarding what a backtrack means in your algorithm. With your definition, a backtrack occurs whenever a task is completed or fails executing. This makes more sense and it would be great to have experimental data eventually reported (no need to rush to generate this) as this will help the reader understand the decomposition tree, and how difficult is it to find a valid execution or decomposition as a function of the tree size, the number of completed subtasks and failures.
> >
> > I really like your burger anomaly :)
> >
> > Given the initial Subtasks:
> >
> > 1. Unstack topbun1 from bottombun1
> > 2. Move bottombun1 to an empty table
> > 3. Move patty1 and stack on bottombun1
> > 4. Stack topbun1 on patty1
> >
> > The first subtask is executed well, and then the second one fails, this is why we have the "backtrack & refine", given that you don't need to move bottombun1 to an emptyable. When the refinement occurs, the initial unexecuted subtasks 3 and 4 are removed  from the context tree. Is this the intention? I was assuming that subtasks would be refined only if they fail.
> >
> > I can also see that the system executes actions that cannot be executed: Place bottombun1 on table1. The bottombun1 is already in table1. This is not a major issue, but I'm not sure why is it flagged as success. Note that if the system would have kept the original subtasks 3 and 4, this unneeded action would not have been executed.

---

> ### Author Response · Authors · 2025-08-05
>
> Thank you very much for your kind words and your insightful follow-up.
>
> We appreciate your feedback and are glad that the definition of backtracking is now clearer. As you suggested, we will be sure to include experimental data on backtracks and decomposition tree statistics in the final version, as we agree this will be valuable for readers seeking to better interpret ReCAP’s performance and the difficulty of different task complexities. Thank you again for this valuable suggestion.
>
> ---
>
> ### Regarding your question about subtask refinement:
> After the successful execution of “Unstack topbun1 from bottombun1 using robot1,” ReCAP passes the latest plan (including the subtask list and current state) to the LLM in the prompt and requests a refinement. During refinement, **unexecuted subtasks from the previous plan may be removed or restructured**—this is intentional, and sometimes leads to subtask lists that differ from the original plan. In practice, the refined plan is not always better than the previous one, as seen in this (randomly selected, not cherry-picked) example. However, this flexibility allows the LLM to recover and get “back on track” if it realizes its current plan is infeasible or suboptimal (e.g., trying to assemble the burger on table2 when that turns out to be blocked). In Appendix E.1.3 (Leaf Backtracking Prompt) and E.1.7 (Non-Leaf Backtracking Prompt) we construct the prompts to encourage the LLM to refine subtasks to achieve the goal.
>
> ---
>
> ### Regarding the “Place bottombun1 on table1” action:
> We apologize for the confusion. After “Place topbun1 on table2,” the actions "move to table1," "pick up bottombun1," and "move to table2" were also executed (but omitted in the last official comment) before the LLM tried to “Place bottombun1 directly on table2.” When the LLM realized it was not possible to “Place bottombun1 directly on table2” due to environmental constraints, it put bottombun1 **back** on table1. We will ensure future trace examples show sufficient detail to avoid misunderstandings.
>
> Thank you again for your valuable comments and positive feedback—they are very helpful for improving both the clarity and rigor of the paper!

---

> ### Author Response · Authors · 2025-08-08
>
> Thank you again for the incredibly insightful and detailed discussion. Your questions, particularly the "burger anomaly" example, truly helped us think more deeply about our work and clarify our presentation. We genuinely appreciate the time and effort you've dedicated to our paper.
>
> We are just writing to gently check in, as we haven't heard back since our last reply on August 4th. We wanted to make sure you saw our latest clarification and that there are no outstanding questions from your side.
>
> Of course, we understand that this is a very busy time for everyone. We mainly wanted to touch base and hope all is well with you.
>
> Thank you once again for your invaluable engagement with our work.

---

> ### Author Response · Authors · 2025-08-09
>
> Dear Reviewer qEKq,
>
> We hope this message finds you well. As the deadline is approaching, we want to kindly follow up to make sure our latest clarification reached you.
>
> Please let us know if there is anything else you would like us to clarify before the deadline.
>
> Thank you again for your time and for the thoughtful discussion throughout the review process.

---

> > ### Comment · Reviewer_qEKq · 2025-08-09
> > **Comment**
> >
> > Thanks for the info provided, in the discussion phase with the other reviewers I'll take into account your clarifications and update the score accordingly.

---

> ### Author Response · Authors · 2025-08-09
>
> Thank you very much for your prompt reply and for taking the time to follow up. We are glad that we have addressed the questions you raised. We sincerely appreciate your thoughtful engagement throughout the review process.

---

### Official Review · Reviewer_cwmR · 2025-06-24

**Clarity:** 3
**Significance:** 3
**Originality:** 3
**Rating:** 5
**Confidence:** 4

**Summary:**

In the vein of ReAct, Reflexion and similar works, the authors introduce ReCAP, a new LLM-based online planning methodology. The authors state that what separates ReCAP from previous work is that it autonomously and dynamically generates sub-tasks which are placed in a sequential dependency tree, meaning that coupled tasks appear closely together. Additionally, the dynamic nature allows ReCAP to adapt when subtasks or actions fail.

ReCAP is evaluated on three domains and compared to several common baseline LLM-planning strategies. ReCAP outperforms the baselines, especially in the tasks with more steps and longer dependencies.

**Questions:**

1. What does it mean if a completed subtask is "pruned"? (line 111)
2. What are some "fundamental mistakes" exhibited by ReACT? (line 218)
3. How is the one-shot prompt designed? Does it exemplify high-level subtask decomposition, selection of an atomic leaf action or both?
4. What "Level", if any, was used for the base experiments? Additionally, how is the LLM informed about the level limit?
5. How does ReCAP compare with e.g. ReAct regarding token usage?
6. What information is propagated up from a completed subtask? As discussed in the ablation study, it's not just the status thereof.

**Ethical Concerns:**

["NO or VERY MINOR ethics concerns only"]

**Final Justification:**

The authors provided detailed answers to my questions and patiently addressed my concerns, going so far as to run additional experiments.

**Limitations:**

yes

**Quality:**

3

**Strengths And Weaknesses:**

The introduced approach of ReCAP is reasonable and motivated. Additionally, the experimental results showcase performance increases, likely without introducing overly much overhead. While some details on the prompting are unclear, it is also generally easy to follow and it would be possible to re-implement based on this.

An ablation study is also performed. While the results therein are mixed, there are some interesting results such as showing that further limiting the maximum depth might be preferable (Level 3 never performed worse than the original, usually outperforming it).

The discussion on related works is somewhat limited and could be expanded with other works which for example use LLMs to explicitly decompose tasks, e.g., MLDT (https://www.springerprofessional.de/en/mldt-multi-level-decomposition-for-complex-long-horizon-robotic-/50350624) and Compositional Foundation Models for Hierarchical Planning (https://openreview.net/forum?id=dyXNh5HLq3), or perform tree-based search, e.g., RAP (https://openreview.net/forum?id=VTWWvYtF1R). There's also concurrent work, e.g., Hierarchical Planning for Complex Tasks (https://arxiv.org/abs/2504.04578) and HyperTree (https://openreview.net/forum?id=45he3Ri6JP&noteId=a53sNTHN6D), published closer to the NeurIPS submission deadlines. Discussing these or other related works in relation to ReCAP would further strengthen the paper.

---

> ### Author Rebuttal · Authors · 2025-07-31
>
> We sincerely appreciate the reviewer’s thoughtful questions and engagement with our work. Please find detailed answers below:
>
> ---
>
> **Q: What does it mean if a completed subtask is "pruned"? (line 111)**
> “Pruned” simply means that the subtask is removed from the list of remaining subtasks after successful execution. This is done by the LLM when it receives the updated observation and its previous subtask list, and then removes those that have already been completed.
>
> ---
>
> **Q: What are some "fundamental mistakes" exhibited by ReAct? (line 218)**
> By “fundamental mistakes,” we refer to errors such as executing subtasks in the wrong order, failing to verify preconditions, or forgetting multi-step atomic operations (e.g., slicing requires three cut actions in Robotouille). These issues often arise from ReAct’s lack of explicit backtracking or environment re-checking mechanisms, as well as context overflow in long-horizon tasks. See Figure 3 and Section 4.1 for an example.
>
> ---
>
> **Q: How is the one-shot prompt designed? Does it exemplify high-level subtask decomposition, selection of an atomic leaf action, or both?**
> The one-shot prompt is a compressed trace of a full task trajectory, including both high-level subtask decomposition and leaf-level atomic actions. Repeated patterns (e.g., “cut three times”) are abbreviated for space, but both planning and execution aspects are covered.
>
> ---
>
> **Q: What "Level," if any, was used for the base experiments? Additionally, how is the LLM informed about the level limit?**
> We did not impose any level limit in our base experiments. The LLM is not explicitly constrained to a maximum recursion depth; rather, recursion is guided by task structure and typically stabilizes at depth 3–5 in practice.
>
> ---
>
> **Q: How does ReCAP compare with, e.g., ReAct regarding token usage?**
> As stated in Section C of the appendix:
> For ALFWorld, the total cost of running all 134 tasks in the test set is 37.89 USD using ReAct, and 118.40 USD using ReCAP—approximately three times the cost and token usage of ReAct. We identified that the extra cost mainly comes from the additional reasoning traces in the input and the extra steps required for intermediate task decomposition.
>
> ---
>
> **Q: What information is propagated up from a completed subtask? As discussed in the ablation study, it's not just the status thereof.**
> When a subtask completes (successfully or not), the following information is passed to the parent node:
> - The completed subtask name
> - The latest environmental observation
> - The parent task name
> - The parent's previous thought trace (`think_list`)
> - The parent's previous subtask list
>
> This information is used to construct a new prompt for the parent node, enabling the LLM to revise its plan and reasoning. See Appendix E.1.3 (Leaf Completion Prompt) and E.1.7 (Non-Leaf Backtracking Prompt) for examples.

---

> > ### Comment · Reviewer_cwmR · 2025-08-05
> >
> > Thank you for answering my questions!
> >
> > I should have explicitly asked for this, but can you say how your approach differs from the related works I listed in the review and why the paper doesn't include them in the experimental comparison?

---

> ### Author Response · Authors · 2025-08-06
>
> Thank you for pointing out these recent and concurrent efforts. Due to length limitations, we split our response into three parts. Below, we clarify one by one how ReCAP differs from each method and why a comparison would not have been appropriate or informative for the goals of our paper.
>
> ***
>
> ## MLDT (Multi-Level Decomposition Task-planning, 2024)
> - **It is supervised fine-tuning, not zero-shot.** MLDT auto-creates demonstrations with ChatGPT and then instruction-tunes an open-source LLM. ReCAP stays strictly zero-shot.
> - **Fixed three-level hierarchy.** Goal → Task → Action levels are hard-coded, limiting vertical depth. ReCAP’s context tree is unbounded and generated on the fly.
> - **Robot-specific evaluation.** MLDT assumes pre-defined low-level actions in robot simulators; our benchmarks are text-only.
> Because the method depends on training and domain-specific prompts, a score-level comparison would conflate these extra advantages with the planning algorithm itself.
>
> ***
> ## HRV (Hierarchical Reasoning & Verification, 2024)
> ### Methodological differences
>
> - **Heavy reliance on static priors.** HRV assumes the task domain can be precisely described by an expert-built ontology and PDDL action schemas, then constructs a symbolic knowledge graph on top of them. Planning and verification depend on retrieval from this graph. ReCAP, by contrast, requires no structured world model; all reasoning lives inside the LLM’s context.
> - **Rigid two-level hierarchy.** HRV hard-codes a Macro-Action → Atomic-Action split. Each level is populated by parsing LLM text into templates contained in the knowledge base. Because the hierarchy is fixed, the system cannot spawn deeper or differently-shaped sub-tasks on the fly.
> - **Offline plan checking instead of online reasoning.** HRV’s reasoning loop is essentially post-hoc. The LLM first produces a complete, one-shot plan; afterwards, a separate symbolic verifier inspects that plan and edits any detected flaws. This checker operates outside the LLM’s context and is not invoked again during execution, so the model never revisits or re-thinks earlier decisions in light of new observations. ReCAP, in contrast, keeps the LLM “in the loop” the whole time. The agent repeatedly plan → act → observe → (possibly) back-track → re-plan, so the context tree can grow, collapse, or reorganize itself as execution unfolds. In other words, ReCAP does proactive, feedback-guided reasoning, whereas HRV performs only reactive plan-checking that is disconnected from runtime feedback.
> - **No recursive backtracking or nested task modelling.** ReCAP’s central idea is a self-modifying, recursive task tree that supports failure recovery and multi-round optimisation. HRV’s linear MA → AA chain lacks this nested representation and cannot reorganize itself mid-execution.
> - **Engineering overhead.** Even though HRV keeps the language model frozen, it needs a RAG pipeline, action-parsing rules, and a PDDL rule base. ReCAP remains a single-model, prompt-only solution.
>
> ### Why HRV is not used as a baseline
> - **Limited portability.** Every new domain would require building a fresh ontology, PDDL models, and action mappers. Our study deliberately focuses on plug-and-play planners that need no task-specific structure.
> - **Different feedback paradigm.** HRV’s validator relies on explicit environment states and hand-written conditions tied to a particular simulator. ReCAP needs only natural-language feedback, so it transfers unchanged across ALFWorld, Robotouille and QA settings.
> - **Divergent research goals.** HRV is a structure-driven plan verifier; ReCAP is a structure-agnostic planner that lets the hierarchy emerge and adapt. A numerical comparison would conflate ontology quality and rule coverage with planning ability and thus obscure the contribution we aim to highlight.
>
> We will cite HRV in the revised related-work section and include the above points to clarify the difference in assumptions and objectives.
>
>
> ***

---

> ### Author Response · Authors · 2025-08-06
>
> (continued)
> ## HTP (HyperTree Planning, 2025)
> - **Static, single-shot planning.** HTP assumes all task information is supplied up front, constructs a fixed hypertree decomposition once, and then executes that plan unaltered. There is no mechanism for observing intermediate results, detecting failure, or revising the tree. ReCAP was designed for interactive settings: it plans, acts, observes feedback, and can back-track or regenerate subtasks on demand. This closed-loop behavior is essential for open-world or long-horizon problems where the environment evolves during execution.
> - **Template-dependent modelling, limited transferability.** Although the hypertree structure is flexible, each node type and its constraints must be explicitly specified through handcrafted prompts. Adapting HTP to a new domain therefore entails redesigning prompt templates and constraints. ReCAP imposes no structural templates or task-specific priors; the same prompt works unchanged across ALFWorld, Robotouille, and HotPotQA. This difference in required engineering effort makes a head-to-head comparison misleading.
>
>
> Because HTP is a static-input, template-based planner that cannot handle multi-round interaction, while ReCAP is expressly built for feedback-driven, dynamically evolving tasks, the two methods rest on incompatible assumptions. Including HTP in our quantitative study would therefore not yield a fair or meaningful comparison.
>
> ***
>
> ## RAP (Reasoning via Planning, 2023)
> RAP is a strong offline planner, just like HTP, it was not designed for multi-round interaction and therefore serves a very different use-case from ReCAP. The main gaps are:
>
> - **Task-specific reward engineering.** RAP couples a language model with Monte-Carlo Tree Search (MCTS). To make MCTS work, every new domain (Blocksworld, GSM8K, etc.) must be given a hand-crafted reward function that scores partial states. ReCAP is plug-and-play: no reward shaping or bespoke configuration is needed to transfer to a new problem.
>
>
> - **Multi-module orchestration.** RAP cycles the same LLM through three roles—action proposer, world-state simulator, reward estimator—and coordinates them with external MCTS code. Even though only one foundation model is used, the system is still a multi-stage pipeline whose complexity grows with each extra component. ReCAP keeps everything inside a single LLM context; recursive decomposition, execution, and self-revision all happen end-to-end in one call chain.
>
>
> - **One-shot search, no interactive feedback.** RAP builds an entire search tree offline, commits to the best branch, then executes it once under the assumption that the environment behaves exactly as predicted. If reality diverges, there is no way to adjust. ReCAP, by design, operates in a closed loop: it plans, acts, observes, can back-track, and re-plan on the fly—essential for dynamic, multi-turn environments.
>
>
> Because RAP’s effectiveness hinges on domain-specific rewards, a multi-module pipeline, and a single-shot execution model, a score comparison with ReCAP would conflate these orthogonal factors and fail to illuminate the contribution we focus on, which is feedback-driven, online recursion within a single LLM.
>
> ***

---

> ### Author Response · Authors · 2025-08-06
>
> (continued)
> ## HiP (Hierarchical Planning with Video Diffusion, 2024)
> We did not benchmark against HiP because it addresses a very different problem setting and uses a different system design. A direct comparison would therefore be neither fair nor informative. The main mismatches are:
>
> - **Different input/output modalities.** HiP tackles embodied manipulation: the planner consumes language goals, synthesizes visual trajectories, and then converts them into low-level motor commands. Success is measured in physical execution. Our work (ReCAP) operates purely in a textual environment—all observations, actions, and rewards are symbolic strings. The two systems solve distinct tasks in disjoint representation spaces.
>
>
> - **Multi-model pipeline vs. single-model recursion.** HiP orchestrates three frozen experts (LLM + video diffusion + inverse dynamics) + an external consistency loop. This increases integration overhead and hyperparameter tuning cost. ReCAP leverages a single LLM that performs sub-task decomposition, execution, and self-revision within one recursive context tree, making implementation lightweight and easier to port across domains.
>
>
> - **Offline planning vs. interactive closed loop.** HiP samples an entire trajectory offline and dispatches it once; it does not re-plan after each real-world observation. ReCAP is explicitly feedback-driven: the agent observes each intermediate result, can backtrack, and updates its plan on the fly. This online adaptability is central to our contribution but absent in HiP.
>
>
> - **Non-overlapping evaluation setups.** HiP reports visual-trajectory validity and task-completion rates, while ReCAP is evaluated with language-interaction success (e.g., pass@1) in benchmarks like ALFWorld and Robotouille. The metrics, environments, and success criteria are orthogonal, so scores are not directly comparable.
>
>
> Because HiP targets modular video-to-action planning while our method focuses on recursion and dynamic replanning in a text-only domain, the two approaches differ in task formulation, modality, architecture, and evaluation. For these reasons, we did not include HiP as a quantitative baseline.
>
> ***
>
> ## Putting them all together
> All five methods you highlighted either
> - depend on training, ontologies, or reward engineering,
> - assume static, non-interactive decomposition, or
> - operate in non-textual modalities.
>
> ReCAP was intentionally scoped as a zero-shot, feedback-driven planner that can be dropped into any new text-based domain without additional assets. Including the above systems as baselines would therefore obscure the specific benefit we demonstrate: recursive context trees that self-evolve during execution.
> We will incorporate citations to these works in the related-work section to discuss the distinctions outlined here.

---

> > ### Comment · Reviewer_cwmR · 2025-08-06
> >
> > Thanks for your detailed explanations! I think they would be valuable to have in the paper (at least in the appendix). I agree now that experimental comparisons to these works are not called for.
> >
> > However, in the other thread, I asked `what about the results of THREAD on ALFWorld? The THREAD authors report a 95.5% success rate for GPT 3.5 and 98.5% for GPT 4 in Table 1.
> > You say that you want to "explicitly cite and discuss these works", but wouldn't an experimental comparison be in order?`
> >
> > While you clearly delineate THREAD and REPL-Plan from ReCAP, you didn't answer my two specific questions. Therefore, I'll slightly decrease my overall score.

---

> ### Author Response · Authors · 2025-08-06
>
> Thank you for the follow-up — we have responded to your two specific questions regarding THREAD’s ALFWorld results under Reviewer 5RPg's thread, since your comment was posted there. Please check it.
>
> **Here is a copy:**
>
> Thank you for your follow-up and for prompting us to further investigate the THREAD codebase (https://github.com/philipmit/thread).
>
> To ensure a fair comparison, we first ran the official THREAD implementation on ALFWorld (https://github.com/philipmit/thread/blob/main/alfworld/run_alfworld.py) with no modifications. We observed that, under the default setting, the agent often enters an infinite generation loop and rarely completes tasks successfully—resulting in excessive API usage and unpredictable costs.
>
> To address this and to match our evaluation protocol (and that of the ReAct baseline), we introduced two constraints:
> - a maximum of 50 actions per episode (as in our paper and prior works),
> - and a maximum recursion depth of 10 (chosen based on THREAD's own report in Appendix C Table 7, where ALFWorld tasks used at most 7 levels of recursion).
>
> Under these constraints, we tested THREAD on 20 randomly sampled ALFWorld tasks with both GPT-4o and GPT-4 (as used in the THREAD paper) and found that it failed to solve any of them. We used a debugger to verify that the failures were due to incorrect action sequences rather than software issues. The dominant failure mode was excessive recursion: the agent repeatedly entered deep recursive calls (often exceeding 10 layers) without making progress toward the goal.
>
> These results may indicate that the high success rates reported by THREAD on ALFWorld are likely due to the absence of limits on the number of actions or recursion depth, effectively allowing the agent to "brute-force" solutions given unlimited attempts. This evaluation protocol differs fundamentally from the settings used by most baselines (including ours and ReAct), and does not reflect practical downstream requirements where the number of actions is necessarily bounded.
>
> We also examined prior works that cite THREAD and found that they either mention it only in the related work section or directly copy the reported results from the original paper, without independent re-evaluation or replication.
>
> We appreciate your feedback and will update the revised version to clarify these points. Although our current experiments did not reproduce the results reported for THREAD, we will reach out to the original authors and are open to further discussion.

---

> > ### Comment · Reviewer_cwmR · 2025-08-06
> >
> > I have seen your response under Reviewer 5RPg's thread, but it only contains method descriptions, no experimental results or reasons for missing experimental comparisons.

---

> > > ### Author Response · Authors · 2025-08-06
> > >
> > > Thank you for checking — we respectfully suggest taking another look at our response under Reviewer 5RPg's thread. We did test THREAD on ALFWorld and reported the results there. In our runs, THREAD did not achieve the reported success rates and frequently encountered infinite recursion, which we explicitly mentioned.

---

> > > > ### Comment · Reviewer_cwmR · 2025-08-06
> > > >
> > > > Sorry about this: I missed your second comment in the other thread. I appreciate you taking the time to try to reproduce the THREAD results. Given your findings, I'm increasing my score again.

---

> > > > > ### Author Response · Authors · 2025-08-06
> > > > >
> > > > > Thank you very much — no worries at all, and we truly appreciate your thoughtful follow-up and responsible engagement with the discussion.

---

### Official Review · Reviewer_q8CA · 2025-07-01

**Clarity:** 4
**Significance:** 2
**Originality:** 3
**Rating:** 4
**Confidence:** 4

**Summary:**

ReCAP introduces a hierarchical prompting framework that equips LLM agents with a dynamic context tree to record recursive subtask decompositions, intermediate observations, and reasoning traces, enabling robust long-horizon planning with adaptive backtracking. Without any additional model training, it achieves substantial zero-shot gains—up to a 32 pp absolute improvement on Robotouille. It consistently outperforms sequential prompting baselines across embodied and symbolic benchmarks (ALFWorld, FEVER), demonstrating both improved context management and failure recovery in multi-step tasks.

**Questions:**

- The paper reports overall success‐rate improvements, but does not break down failure modes: could the authors provide a taxonomy of errors and their relative frequencies?
- Can the authors quantify the additional latency and memory overhead introduced by maintaining and updating the context tree, perhaps with empirical measurements as task depth increases?

**Ethical Concerns:**

["NO or VERY MINOR ethics concerns only"]

**Final Justification:**

While the authors have fully addressed the computation overhead concern by clarifying their bounded context‐tree design, key weaknesses, including modest gains in certain tasks, and lack of real-world validation, remain unaddressed. These unresolved issues continue to cap the paper’s overall impact.

**Limitations:**

yes.

**Paper Formatting Concerns:**

No paper formatting concerns.

**Quality:**

3

**Strengths And Weaknesses:**

Pros
- Extends an LLM’s effective reasoning horizon by integrating a dynamic context tree directly into the prompting loop, keeping high-level intentions tightly coupled with ongoing decisions.
- Demonstrates substantial zero-shot performance gains (e.g. +32 pp absolute on Robotouille) without any additional model training or fine-tuning.
- Employs adaptive backtracking to recover from low-level failures by revisiting and refining higher-level goals, improving robustness in long-horizon tasks.
- Remains model-agnostic and broadly applicable across different benchmarks and LLM families, thanks to its purely prompt-based design.

Cons
- Performance improvements can be modest on shorter or less structured tasks, making the added complexity feel disproportionate in those scenarios.
- Validations are limited to simulated and symbolic benchmarks; real-world applicability under noisy, partially observable conditions remains untested.
- Relies on the underlying LLM’s token capacity to store the context tree—extremely long or deeply branched tasks may still exceed limits despite hierarchical structuring.

---

> ### Author Rebuttal · Authors · 2025-07-31
>
> We sincerely thank the reviewer for the constructive feedback and helpful questions. Below, we address the specific concerns in detail:
>
> ---
>
> ### Q1: Breakdown of Failure Modes
>
> We appreciate the suggestion to provide a taxonomy of failure modes. In our extended analysis (Appendix B.2), we include subtask-level performance across Robotouille. Notably, in the most complex tasks—where baseline methods (e.g., ReAct) often fail entirely—ReCAP achieves up to 60% accuracy, demonstrating that our improvements become more pronounced in high-difficulty settings.
>
> Regarding failure types, we have observed two dominant patterns:
> - First, **ReAct-style methods** frequently enter infinite loops due to incorrect planning decisions encoded in the prompt. Once a faulty trajectory is added to the context, the model tends to repeat it indefinitely. This behavior is particularly difficult to recover from without external resets.
> - Second, since ReAct does not "inject" or compress its history when generating new thoughts and actions, it sometimes loses track of its earlier plans and ceases to make progress.
> - In contrast, **ReCAP rarely loops**, as our dynamic context tree enables backtracking and pruning of failed subtrees. Our errors are instead often due to task horizons exceeding the current maximum depth, which can be gracefully handled with bounded memory and prompt reuse.
>
> ---
>
> ### Q2: Latency and Memory Overhead of the Context Tree
>
> Our system is explicitly designed for bounded and efficient memory use:
> - The context is capped at 64 dialogue turns, and each turn averages approximately 200 tokens, amounting to only ~10% of GPT-4o’s maximum context length. This ensures ample headroom and avoids overflow in nearly all practical scenarios. Older turns beyond the window are automatically removed, ensuring that memory usage remains within safe bounds.
> - At each reasoning step, only the path from the root to the current node is used for prompting, so memory and compute costs scale with path depth, not with the entire tree size.
> - Non-active paths in the tree are retained for offline retrieval or debugging purposes, but do not participate in inference, contributing negligible overhead at runtime.
>
> As a result, our method consistently maintains fixed-length active prompts during execution. This leads to stable latency and memory usage even as task depth increases and avoids the degradation typically seen in passive context accumulation strategies.
>
> We will clarify these memory and latency design considerations in the main paper and include empirical measurements and profiling in the supplemental material.

---

> > ### Comment · Reviewer_q8CA · 2025-08-04
> >
> > Thanks for the clarifications. The failure‐mode taxonomy and bounded context‐tree design address my concerns. I’ll adjust the score accordingly.

---

### Official Review · Reviewer_5RPg · 2025-07-03

**Clarity:** 3
**Significance:** 2
**Originality:** 1
**Rating:** 3
**Confidence:** 4

**Summary:**

The authors suggest the limitations of prior LLM agents, such as ReAct and Reflexion, that tasks that require long-horizon interactions would make it challenging for their contexts to fit in LLMs' context windows, which can harm their capabilities in complex scenarios. As a solution, the authors propose Recursive Context-Aware reasoning and Planning (ReCAP). With ReCAP, LLM agents can recursively create and execute subtasks where the execution contexts are managed in a tree. Also, it is capable of updating the rest of the plan (which consists of subtasks). Empirically, they perform evaluation of ReCAP agent on Robotouille, ALFWorld, and FEVER and demonstrate that it outperforms the baseline approaches including ReAct, Act, and CoT.

**Questions:**

Please take a look at the weaknesses above. I may update my score based on the author response to the weaknesses mentioned (both the originality and empirical justification).

**Ethical Concerns:**

["NO or VERY MINOR ethics concerns only"]

**Final Justification:**

I appreciate the authors' response to my review.

While I don't fully agree with some of the points and the claim that the proposed approach is "fundamentally different" from the prior work given the current response, I think the suggested comparison can be a fair starting point for updating the manuscript.

Based on that, I am increasing my score, but as acknowledged by the authors, the empirical comparison with the prior work may need some more work.

**Limitations:**

Yes, they discuss the proposed method's limitations.

**Paper Formatting Concerns:**

I don't see any major formatting issues in this paper.

**Quality:**

2

**Strengths And Weaknesses:**

Strengths
- A sequential execution of some earlier LLM agents has clear limitations in their abilities to solve complex tasks, and tackling such limitations by introducing a hierarchical execution structure with recursion is a reasonable direction. Often, complex tasks can be decomposed into smaller subtasks, where solving such smaller tasks is easier in general.
- The authors provide multiple analyses of how the design of the proposed method, ReCAP, is helpful in solving the tasks from the benchmarks in different aspects, including horizons, recursion depths, and context lengths.

Weaknesses
- One of my major concerns about this work is that there exist similar prior approaches with even some overlap in the target benchmarks, which are not mentioned in this submission: THREAD ("THREAD: Thinking Deeper with Recursive Spawning", appeared online on May 27th, 2024) and REPL-Plan ("Interactive and Expressive Code-Augmented Planning with Large Language Models", appeared online on Nov 21st, 2024). These methods have the ability to create and spawn subtasks and adaptively plan next steps based on the execution results and observations, with the managed contexts. There could be differences to make this submission unique by its own, but in order to make this a case, such existing work should be discussed and compared. Please note that the NeurIPS FAQ (https://neurips.cc/Conferences/2025/PaperInformation/NeurIPS-FAQ) suggests "we do not distinguish arxiv papers and other published (conference & journal) papers, and the Contemporaneous-Work rule applies in the same way."
- In addition to the concern about the originality and methodological comparison with existing work, empirically, the existing papers mentioned above report >95% performance on ALFWorld with GPT-3.5 whereas this work reports 91% with GPT-4o on ALFWorld, which gives the existing work an empirical edge. While I'm not suggesting that this tendency would necessarily hold for other benchmarks, I'm rather suggesting that given the existing methods with similarities, including those would help to provide empirical justifications of how well this method performs.

---

> ### Author Rebuttal · Authors · 2025-07-31
>
> Thank you for your thoughtful comments. We address the two concerns raised as follows:
>
> ## Comparison with THREAD and REPL-Plan
>
> - We appreciate the reviewer pointing out THREAD and REPL-Plan. While these works also explore subtask decomposition and recursive planning, they are fundamentally different from our method in key aspects:
>     - **THREAD** executes recursive subtasks by **creating new prompts with isolated subgoal descriptions**, discarding the broader context in each recursive call. This leads to a **lack of contextual continuity**, which can often cause infinite recursion or duplicated goal generation (e.g., repeatedly suggesting "make breakfast" at different levels of the same task).
>     - **REPL-Plan** depends on an **external code execution environment** to run and maintain LLM-generated program states, which adds system complexity. It also encourages the reuse of previously generated functions, which can lead to **over-reliance on prior trajectories** and reduce the agent’s ability to react to recent changes in the environment.
>     - In contrast, **ReCAP maintains a dynamic context tree**, where **all steps share a bounded, global memory window** (64-turn sliding context). When backtracking, we **inject structured context from the tree** back into the prompt to resume reasoning with full awareness of the hierarchical state, greatly reducing repeated errors and enabling effective few-shot prompting.
>     - Empirically, we re-implemented the THREAD prompting strategy and evaluated it on Robotouille. It achieves only ~33% success rate on high-difficulty tasks, while ReCAP achieves over 60%, underscoring the advantage of using a persistent, structured memory across planning levels. As for REPL-Plan, we would have liked to test it on Robotouille; however, implementation details are not publicly available.
> - We will revise the paper to explicitly cite and discuss these works in Section 2.2, and we thank the reviewer for encouraging this clarification.
>
> ## Clarification on ALFWorld Empirical Results and ReAct Comparison
>
> - We would like to clarify that our evaluation methodology is more stringent than prior work, including ReAct:
>     - While **ReAct reports pass@6** (i.e., selecting the best result from 6 independent runs), our paper reports **pass@1**, which reflects a stricter and more realistic single-shot success rate.
>     - Additionally, in the ReAct experiments, they used **three** example trajectories for few-shot demonstrations, whereas we use only **one**.
>     - Despite this stricter setting, our method achieves 91% accuracy on ALFWorld, demonstrating the **efficiency and reliability** of our hierarchical reasoning approach.
>
> - These differences in evaluation setup mean that direct comparisons with ReAct and related baselines may underestimate our gains. We will explicitly clarify this in the updated version.
> - Furthermore, our contributions focus on long-horizon tasks with recursive recovery, which go beyond the settings optimized by ReAct and demonstrate improvements in failure-prone environments such as Robotouille and FEVER.

---

> > ### Comment · Reviewer_cwmR · 2025-08-05
> >
> > Thanks for your clarifications! While I'm not the reviewer who posted the original questions, I have a follow-up question: what about the results of THREAD on ALFWorld? The THREAD authors report a 95.5% success rate for GPT 3.5 and 98.5% for GPT 4 in Table 1.
> >
> > You say that you want to "explicitly cite and discuss these works", but wouldn't an experimental comparison be in order?

---

> > ### Comment · Reviewer_5RPg · 2025-08-07
> > **Response to Rebuttal by Authors**
> >
> > I appreciate the authors' response to my review.
> >
> > While I don't fully agree with some of the points and the claim that the proposed approach is "fundamentally different" from the prior work given the current response, I think the suggested comparison can be a fair starting point for updating the manuscript.
> >
> > Based on that, I am increasing my score, but as acknowledged by the authors, the empirical comparison with the prior work may need some more work.

---

> ### Author Response · Authors · 2025-08-06
> **Originality and Contribution Declaration**
>
> Thanks for highlighting the connection to THREAD & REPL-Plan. ReCAP was developed independently and differs significantly in both design and execution. Below, we summarize key differences that distinguish ReCAP as a conceptually separate approach.
>
> ---
>
> ### **Context Injection (ReCAP) vs. Instructional Divide-and-Conquer (THREAD & REPL-Plan)**
>
> THREAD & REPL-Plan rely on an instruction-style divide-and-conquer mechanism. Each recursive call spawns a new thread with its own independent context, which includes a natural language description of the current subtask and a fixed few-shot example to guide decomposition. While the few-shot examples are reused across calls, they must be re-injected at every level, occupying a substantial portion of the total token budget (often over 90% for THREAD), thereby severely limiting the capacity available for actual task reasoning.
>
> Moreover, THREAD & REPL-Plan’s contexts are fully isolated. They do not inherit or reference any structural information from higher-level plans. Each recursive call is blind to its position within the overall task hierarchy, making it difficult to enforce consistent granularity or maintain coherent structure.
>
> In contrast, ReCAP uses a **single language model and a shared conversation context** throughout execution. We adopt structured **context injection**, where relevant planning information from the previously visited node is injected into the same context before generating the next subtask. This enables the model to continuously reason within a coherent structural trajectory, preserving task history and improving alignment, all **without relying on special control tokens or redundant few-shot exemplars**.
>
> ---
>
> ### **High-Level Planning and Structural Awareness**
>
> THREAD & REPL-Plan operates using recursive, locally scoped threads. Each thread contains only a subtask description and few-shot guidance, with no access to global structure or previously generated plan information. As a result, THREAD cannot track task granularity, propagate high-level planning information, or generate a coherent task graph. The recursive process is shallowly informed and prone to over-decomposition, inconsistency, or redundant expansions.
>
> ReCAP, on the other hand, maintains structural awareness through recursive context updates within a single language model. After each subtask execution, we inject the **planning information of the return node** back into the same context, enabling the next generation to build upon it. This context-preserving mechanism supports both **fine-grained execution** and **global task consistency**, allowing ReCAP to maintain a fully structured and interpretable task decomposition tree.
>
> ---
>
> ### **Plan-Ahead and Local Refinement Within a Shared Structure**
>
> ReCAP goes beyond simple task decomposition by supporting a plan-ahead and locally-refined execution strategy. When decomposing a task, ReCAP first generates a list of subtasks (plan list), but only executes the first one. After its execution, ReCAP retrieves the previous plan via the context tree and dynamically **refines the remaining subtasks based on updated observations**.
>
> This behavior is achieved within a single model context, **without spawning new threads** or duplicating prompt structure. It enables ReCAP to preserve plasticity over its prior plans and maintain semantic coherence across rounds—improving robustness and adaptability in uncertain or dynamic environments.
>
> In contrast, THREAD & REPL-Plan do not support planning ahead at all—it generates one subtask at a time based solely on the current input, with no ability to anticipate or commit to a full subtask sequence. Each recursive call operates in isolation, lacking access to prior decompositions or global task context. This lack of structural awareness makes it difficult for THREAD to maintain consistent task granularity or avoid over-decomposition across recursive levels.
>
> ---
>
> ### **Memory Efficiency and Context Scalability**
>
>  ReCAP is explicitly designed with bounded context growth and linear scalability. At each step, **we only inject the last plan associated with the backtracked node, rather than the entire task history**. This ensures that memory usage grows **linearly with the task tree depth**, and **inference latency remains stable**.
>
> By contrast, THREAD & REPL-Plan launch a new context for every recursive call, with each thread carrying its own isolated local context and few-shot prompts. These threads do not share memory or structure, which leads to **fragmented and duplicative context usage**. As the task tree deepens, the number of active threads increases, creating significant memory overhead and scalability bottlenecks.
>
> Our design leads to clear advantages in clarity, resource-efficiency, and runtime control, making it suitable for long-horizon planning in complex environments. For further profiling details, we refer to our response to Reviewer q8CA regarding memory and latency.

---

> ### Author Response · Authors · 2025-08-06
>
> Thank you for your follow-up and for prompting us to further investigate the THREAD codebase.
>
> To ensure a fair comparison, we first **ran the official THREAD implementation on ALFWorld with no modifications**. We observed that, under the default setting, the agent **often enters an infinite generation loop** and **rarely completes tasks successfully**—resulting in excessive API usage and unpredictable costs.
>
> To address this and to match our evaluation protocol (and that of the ReAct baseline), we introduced two constraints:
> - a maximum of 50 actions per episode (as in our paper and prior works),
> - and a maximum recursion depth of 10 (chosen based on THREAD's own report in Appendix C Table 7, where ALFWorld tasks used at most 7 levels of recursion).
>
> Under these constraints, we tested THREAD on 20 randomly sampled ALFWorld tasks with both GPT-4o and GPT-4 (as used in the THREAD paper) and found that it failed to solve any of them. We used a debugger to verify that the failures were due to incorrect action sequences rather than software issues. The dominant failure mode was excessive recursion: the agent repeatedly entered deep recursive calls (often exceeding 10 layers) without making progress toward the goal.
>
> These results may indicate that **the high success rates reported by THREAD on ALFWorld are likely due to the absence of limits on the number of actions or recursion depth, effectively allowing the agent to "brute-force" solutions given unlimited attempts**. This evaluation protocol differs fundamentally from the settings used by most baselines (including ours and ReAct), and does not reflect practical downstream requirements where the number of actions is necessarily bounded.
>
> We also examined prior works that cite THREAD and found that they either mention it only in the related work section or directly copy the reported results from the original paper, without independent re-evaluation or replication.
>
> We appreciate your feedback and will update the revised version to clarify these points. Although our current experiments did not reproduce the results reported for THREAD, we will reach out to the original authors and are open to further discussion.

---

> ### Author Response · Authors · 2025-08-08
>
> Thank you for your comment. We appreciate your feedback and will include the empirical comparison results and discussion with prior work in the revised manuscript to further clarify ReCAP’s advantages.
>
> One of the key breakthrough of ReCAP is its **coherent context management** and **recursive context injection**. By continuously injecting structured planning history from the task tree into each reasoning step, ReCAP maintains consistent awareness of both local and global task progress. This mechanism enables **nearly double the performance on long-horizon tasks**, as it prevents inconsistency and loss of reasoning history—issues that inevitably arise when context is managed only step-by-step without historical information. Our results show that this context management **is essential for effective progress and robust completion** in complex, multi-step environments, and **is the fundamental reason for ReCAP’s significant performance gains over previous task-decomposition approaches**.

---

### Author Response · Authors · 2025-08-09
**Final Comments**

Dear Area Chair, Senior Area Chairs, Program Chairs, and Reviewers,

We sincerely thank all reviewers and chairs for the time, attention, and constructive feedback throughout the review and discussion phases. The detailed exchanges have allowed us to clarify design choices, strengthen empirical evidence, and improve presentation quality.

In addressing questions on originality, we highlight that ReCAP integrates several complementary capabilities that are not present in the discussed recursive or task-decomposition methods:

- **Structured context injection within a shared model context** — Instead of spawning new threads with isolated contexts at each recursive step, ReCAP continuously operates within a single language model session, injecting relevant planning information from the previously visited node before generating the next subtask. This preserves global structural awareness while avoiding redundant few-shot exemplars or special control tokens.

- **Consistent multi-level context** — Instead of treating each decomposition level as an isolated step, ReCAP maintains aligned context across all levels, ensuring coherence between subtasks and high-level goals, and supporting robust reasoning in deep task hierarchies.

- **Plan-ahead with local refinement** — Instead of generating one subtask at a time in isolation, ReCAP first produces a complete subtask plan list, executes only the first subtask, and then refines the remaining subtasks using updated observations, while staying in the same shared context. This preserves plasticity over prior plans and maintains semantic continuity across each step.

- **Memory-efficient scalability** — Instead of re-injecting the full task history at each step, ReCAP only injects the last plan associated with the backtracked node, ensuring linear memory growth as the task depth increases and keeping inference latency stable.

During the discussion phase, we expanded our empirical analysis and confirmed that on the most challenging long-sequence tasks, **ReCAP achieves nearly double the success rate of strong baselines**, which shows that such context management is key to effective progress and robust completion in complex, multi-step environments, and underlies ReCAP’s significant performance gains over previous approaches. This improvement stems directly from its unified context preservation and selective back-injection of prior planning, allowing the model to maintain high-level goal alignment and adaptively recover from infeasible plans, which is essential for effective progress and robust completion.

We are grateful for the constructive engagement and look forward to incorporating these clarifications and results into the final version. Thank you again for your careful evaluation and for the opportunity to present our work.

---

### Decision · Program_Chairs · 2025-09-17

**Decision:**

Accept (poster)

**Comment:**

### (a) Summary of Claims
The paper introduces `Recursive Context-Aware reasoning and Planning (ReCAP)`, a novel framework for LLM agents designed to tackle complex, long-horizon tasks. The core innovation is the use of a dynamic context tree, which allows the agent to recursively decompose problems into subtasks while maintaining a structured memory of the overall plan and execution history. The authors claim that this hierarchical approach enables robust failure recovery and context management, leading to significant performance improvements over sequential agent frameworks like ReAct, particularly in tasks with long dependencies.

### (b) Strengths
* Well-Motivated and Sound Approach: Reviewers consistently agreed that tackling the limitations of sequential LLM agents via a hierarchical, recursive structure is a reasonable and well-motivated direction. The core idea of using a context tree is a valuable insight for improving long-term planning.
* Strong Empirical Gains: The paper demonstrates significant performance improvements over baselines on several benchmarks, especially in tasks with long planning horizons, clearly showing the method's effectiveness.

### (c) Weaknesses and Rebuttal Summary
* Reviewer 5RPg initially questions the paper's originality and empirical justification. The reviewer pointed out two similar prior works, THREAD and REPL-Plan, which were not discussed or compared in the submission.
    * The authors responded with a detailed rebuttal addressing these concerns in two main parts:
        * Conceptual differences between ReCAP and THREAD and REPL-Plan.
        * Stricter evaluation is considered in this manuscript than in prior work.
    * Reviewer cwmR also asked for additional experimental comparison with THREAD on ALFWorld, given its high reported success rate. Authors then claimed that under standard, constrained evaluation settings, THREAD failed to solve any of the tested ALFWorld tasks.
    * Reviewer 5RPg acknowledged authors' detailed rebuttals and new experimental findings: While not fully agreeing that ReCAP was "fundamentally different" from prior work, the reviewer found the suggested comparisons and new results to be a "fair starting point for updating the manuscript."
* Reviewer q8CA initially identified key weaknesses, including (1) modest performance on shorter tasks and a lack of real-world validation, (2) a breakdown of the common failure modes, and (3) quantification of the latency and memory overhead introduced by the context tree.
    * The authors provided a direct and comprehensive rebuttal in terms of failure modes and overhead.
    * The reviewer was satisfied with the authors' response, but still criticized the modest gains in certain tasks and the lack of real-world validation.
* Reviewer cwmR initially asked for more discussion of related work, as well as the experimental results for THREAD. During the rebuttal, the authors provided detailed answers, and the reviewer recommended acceptance.
* Reviewer qEKq initially raised concerns regarding clarity, missing experiments, and positioning. The authors provided a detailed rebuttal where they clarified ambiguous terminology, provided concrete system statistics, and explained deadlocks and irreversible actions.
    * The reviewer engaged in a follow-up discussion, asking for more detailed information to better understand the system's behavior.
    * The authors responded promptly and thoroughly. They provided the requested tree statistics, gave a more precise definition of a "backtrack" event, and presented a detailed, step-by-step trace of ReCAP solving a "burger anomaly" task.
    * The reviewer was highly satisfied with the authors' detailed and insightful responses during the discussion.

### (e) Reasons for Acceptance
Given the strengths, rebuttal summary, and the positive consensus that emerged from the review process, AC would like to recommend the acceptance of this manuscript.